# Multi-Level Spatial Embedding Sharing for Enhanced Online Trajectory-User Linking

## Abstract

Trajectory-User Linking (TUL) is a critical task in mobility applications that links unlabeled spatial trajectories to the users or entities that generated them. In these applications, data often arrives as a continuous stream and may experience distributional shifts over time. While adapting TUL models via online learning could address these challenges, this approach remains unexplored in current research. Our work bridges this gap by conducting comprehensive evaluations of common TUL techniques in an online learning context. To improve the performance of existing TUL techniques in this setting, we further introduce Multi-Level Spatial Embedding Sharing (MiLES), an embedding approach that adapts multi-level spatial representation to the online TUL setting. MiLES operates by partially sharing embeddings for locations within neighborhoods of multiple size levels. This design enables generalization of knowledge within neighborhoods, while maintaining fine-grained discrimination through more location-specific representations. MiLES also significantly reduces the number of embedding parameters leading to lower memory usage and more computationally efficient model updates. We further incorporate learnable weighting parameters for each embedding level, allowing the model to learn the influence of different levels during training. Our experimental results on several real-world datasets show that integrating MiLES into state-of-the-art TUL models significantly improves their performance in online learning scenarios, yielding relative gains in top-1 accuracy of up to 24%. To demonstrate its general applicability, we also evaluate MiLES on the task of destination prediction, where it also provides consistent performance improvements, confirming its value as a domain-general embedding technique. Our code is available at `https://anonymous.4open.science/r/MiLES-3D20`.

## 1 Introduction

The proliferation of location-enabled devices has produced vast amounts of mobility data, fueling advances in various machine learning tasks (Rehman et al., 2015; Zheng, 2015). One such task is Trajectory-User Linking (TUL), introduced by Gao et al. (2017). TUL aims to associate chronologically ordered sequences of check-ins at visited locations or points of interest (POIs), so-called trajectories, to the users or entities that generated them. The TUL task has numerous real-world applications, including disease control, law enforcement, ride-sharing, and location-based recommendations (Hao et al., 2020; Gao et al., 2017). Traditional approaches for TUL have demonstrated impressive performance using recurrent neural networks and, more recently, transformer-based architectures.

However, these methods are designed for batch learning, where all training data is available beforehand, and to the best of our knowledge, no prior work has investigated TUL in online learning settings. In many practical scenarios, such as the deployment of a new location-based service, initial data is often limited, and data instead arrives as a continuous stream. In addition, mobility data streams are subject to temporal dependencies and distributional shifts, commonly known as concept drift, caused by factors such as changed traffic routing, evolving user behavior or the emergence of new POIs. External factors, such as the COVID-19 pandemic, are another potential cause for drifts (Borkowski et al., 2021), demonstrating the necessity for models that can adapt dynamically. Online learning offers a promising solution for this by allowing models to incrementally update their knowledge with new trajectory data. In contrast to traditional batch

learning, online models process data sequentially as it arrives, enabling adaptation to changing mobility patterns. While most publicly available trajectory datasets are small enough to allow for periodic batch retraining, online learning remains attractive as a principled framework for non-stationary environments. Examples include privacy-sensitive applications, where discarding trajectory data immediately after each model update — rather than retaining it for periodic retraining — reduces the risk of data breaches, and on-device deployment, where hardware constraints make full retraining costly.

According to the definition of online machine learning by Bifet et al. (2010), a model operating in such an environment must be able to:

**R1**: process a single instance at a time,

**R2**: process each instance in a limited amount of time,

**R3**: use a limited amount of memory,

**R4**: predict at any time,

**R5**: adapt to changes in the data distribution.

Most existing TUL approaches can be adapted for online TUL, which we formalize in Equation 2, by updating models incrementally. However, their ability to predict at any time and adapt to distributional shifts can be hindered by their embedding strategy. To embed check-in locations, most models rely on lookup tables containing separate learnable embedding vectors for each POI. For any given trajectory, only the embeddings for the visited locations receive non-zero gradients with this approach. However, given the large number of unique POIs in most TUL applications, most locations receive relatively few check-ins (Chen et al., 2022). This results in the gradients for POI-based embeddings often suffering from a high degree of sparsity. While in batch learning, this sparsity can be partially mitigated by training on the same trajectories for multiple epochs, it remains a limiting factor, particularly in online learning settings where each trajectory is observed only once and quick adaptation to new data is crucial (**R5**).

A potential solution is to share embeddings across multiple locations to reduce gradient sparsity, as was explored in previous works on batch-learning TUL (Yang et al., 2021; Alsaeed et al., 2023). Sharing embeddings based on spatial proximity allows models to generalize knowledge across locations, improving adaptation for rarely visited POIs. However, there is a trade-off: larger neighborhood sizes reduce the gradient sparsity, but also reduce the specificity of embeddings. This means that while broad embedding sharing may be beneficial at the start of training or following a concept drift, it likely degrades the model's ability to learn fine-grained location details.

To address this issue, we propose M̲ulti-L̲evel Spatial E̲mbedding S̲haring (MiLES), an embedding approach that adapts and extends the principle of multi-scale spatial representations, with particular benefits for online TUL. MiLES partitions embeddings into multiple segments, each shared across neighborhoods of different sizes. This design enables a wide spectrum of representations with varying levels of specificity and gradient density, while also reducing the overall parameter count and improving the computational efficiency of model updates compared to a purely POI-based approach without shared embeddings. MiLES additionally features a learnable weighting mechanism that adjusts the contribution of each embedding level throughout the online learning process, allowing the model to emphasize broader or finer-grained spatial information as needed. While MiLES also improves performance in a conventional batch learning setting, we show in Section 6 that its benefits are consistently more pronounced in the online regime, where each trajectory is seen only once and gradient sparsity is therefore more severe.

To evaluate the effectiveness of MiLES, we first conduct a comprehensive assessment of existing state-of-the-art TUL techniques in an online learning setting. We then integrate MiLES into these techniques and demonstrate its significant performance improvements. Lastly, we perform ablation studies to analyze the contribution of each MiLES component and gain deeper insights into its functionality.

While our primary focus is on TUL, the modular and task-agnostic design of MiLES makes it applicable to other spatial machine learning tasks that benefit from multi-scale representations. To demonstrate this

versatility, we successfully apply MiLES to destination prediction, showing its broader utility beyond TUL (see Section 7).

In summary, our main contributions are: (i) the first systematic evaluation of state-of-the-art TUL models in an online learning setting; (ii) MiLES, a multi-level spatial embedding sharing approach that adapts grid-based spatial sharing to the online TUL setting, reducing gradient sparsity through structured partitioning and learnable level weighting; (iii) an empirical demonstration that MiLES improves top-1 accuracy by up to 24% while reducing parameter counts; and (iv) a demonstration of MiLES' generality via the destination prediction task.

## 2 Preliminaries

In the following we will introduce the basic concepts underlying online trajectory user linking:

**Definition 2.1** (Check-In). A check-in is a tuple $c = (t, l)$ containing a timestamp $t$ and the coordinates $l$ of a visited location of form (latitude, longitude). Check-ins often additionally contain a unique identifier $p \in \mathbb{P}$ for the Point Of Interest (POI) at $l$, where $\mathbb{P}$ is a finite set of POI identifiers.

**Definition 2.2** (Trajectory). A trajectory $T$ is a chronologically ordered sequence of $n$ check-ins, $T = [c_0, c_1, \ldots, c_n]$. Each trajectory is generated by a single user $u$ from a set of users $\mathbb{U}$. A pair $(T, u)$ is a *linked trajectory*. When the generating user is unknown, the trajectory $T$ is considered *unlinked*.

**Definition 2.3** (Trajectory User Linking). The task of trajectory-user linking is to learn a function $f$ on a training set of linked trajectories $\mathbb{D} = \{(T_0, u_0), \ldots, (T_n, u_n)\}$, that correctly assigns a user label to each unlinked trajectory in a test set $\mathbb{D}^{(\text{test})}$. Formally, the objective is to minimize the predictive loss:

$$\min_{\boldsymbol{\theta}} \sum_{(T_i, u_i) \in \mathbb{D}^{(\text{test})}} L(f(T_i; \boldsymbol{\theta}, \mathbb{D}^{(\text{train})}), u_i), \tag{1}$$

where $L$ is a loss function quantifying the predictive error, and $\boldsymbol{\theta}$ are the model parameters to be optimized. In the batch setting, this objective reflects training on $\mathbb{D}^{(\text{train})}$ and evaluating on a held-out test set.

In an online setting, where samples arrive sequentially and cannot be stored indefinitely, TUL is more effectively evaluated under the *prequential* or *interleaved test-then-train* scheme (Bifet et al., 2010), where each incoming sample is first used to test the model and then for updating it. Given a stream of pairs $\mathbb{S} = \{(T_0, u_0), \ldots, (T_m, u_m)\}$, the objective becomes:

$$\min_{\boldsymbol{\theta}_0, \ldots, \boldsymbol{\theta}_{m-1}} \sum_{i=1}^{m-1} L(f(T_i; \boldsymbol{\theta}_{i-1}, \mathbb{S}_{:,:i-1}), u_i), \tag{2}$$

where $\boldsymbol{\theta}_{i-1}$ are the model parameters at time $i-1$, and $\mathbb{S}_{:,:i-1}$ denotes all previously seen pairs. Therefore, the parameters at each step of the training process contribute equally to the performance of an online TUL model, making it more susceptible to the embedding sparsity issue mentioned in section 1.

Following the standard supervised online learning protocol (Bifet et al., 2010), we assume that the true user label $u_i$ becomes available after each prediction, allowing the model to update on $(T_i, u_i)$ before processing the next instance. This assumption is common in the online learning literature and is reasonable in applications such as ride-hailing, where the system predicts which user is most likely to service a requested route and observes the true assignment after completion. We acknowledge that settings with delayed or incomplete feedback may be more realistic for certain applications, and consider the extension of our approach to such settings a promising direction for future work.

## 3 Related Work

While trajectory user linking itself is a relatively recent task, it builds upon established methods from adjacent fields. Early approaches adapted the longest common subsequence (LCS) algorithm (Ying et al., 2011) to predict user labels by finding the longest shared sub-trajectory between unlabeled and known trajectories.

Similarly, bag-of-words representations, which encode trajectories based on POI visit frequencies (Mikolov et al., 2013), enable the application of conventional classification methods such as linear discriminant analysis and support vector machines to the TUL problem.

More recent approaches, starting with TUL via Embedding and RNN (TULER) (Gao et al., 2017), use a lookup-table embedding scheme that preserves the temporal order of check-ins. Gao et al. (2017) introduced three TULER variants (TULER-G, TULER-L and BiTULER) combining this embedding approach with GRU (Cho et al., 2014), LSTM or bidirectional LSTM networks (Hochreiter & Schmidhuber, 1997). In subsequent works, various extensions of TULER were proposed, including TULVAE (Zhou et al., 2018) which combines an LSTM classifier with a variational autoencoder, and DeepTUL (Miao et al., 2020) which extends TULER with a historical attention module based on user IDs of previous check-ins sharing the same locations and time-slots.

With their advancement in other machine learning disciplines, newer studies on TUL have increasingly focused on transformer-based architectures. The T3S model (Yang et al., 2021) combines transformer and LSTM encoders to encode trajectories before classification. The purely transformer-based TULHOR (Alsaeed et al., 2023) embeds check-ins using hexagonal grids and supplements these with conventional POI embeddings.

Notably, several of these and other works have explored grid-based or multi-scale spatial encodings for trajectory and geographic tasks. T3S embeds locations by mapping them to grid cells, but uses only a single grid with fixed resolution, limiting its embeddings due to the trade-off between gradient density and specificity. TULHOR similarly relies on a single hexagonal grid level. Outside of TUL, Hu et al. (2022) represent locations with geohashes at varying resolutions and combine them via feature hashing for time-of-arrival estimation, demonstrating the utility of capturing spatial information at multiple granularities. More broadly, Space2Vec (Mai et al., 2019) encodes geographic coordinates using multi-scale sinusoidal functions inspired by biological grid cells. Similarly, fourier features (Tancik et al., 2020) leverage sinusoids of varying frequencies to map coordinates into higher-dimensional spaces. While these continuous encoding approaches operate at the coordinate level, MiLES shares their underlying principle of multi-scale spatial representation but applies it within the discrete lookup-table framework typical of TUL architectures. Specifically, MiLES contributes a learnable weighting mechanism that allows the model to adaptively focus on different scales and a structured partitioning of the embedding dimensions. This is detailed in Section 4 and is motivated by the severe gradient sparsity constraint in online learning settings, which is not addressed by any of the aforementioned approaches.

Other notable TUL models have employed approaches include MainTUL (Chen et al., 2022), which combines transformers and RNNs through mutual distillation, and TGAN (Zhou et al., 2021c), which uses GANs for data augmentation. Further TUL models include GNNTUL (Zhou et al., 2021a) and AttnTUL (Chen et al., 2024), which process trajectories using graph neural networks, the self-supervised learning approach SML (Zhou et al., 2021b), as well as the Siamese neural network TULSN (Yu et al., 2020).

Outside of TUL, Fourier features (Tancik et al., 2020) are commonly used for applications like Neural Radiance Fields (Mildenhall et al., 2020), mapping coordinates into a higher-dimensional space using multiple sinusoidal functions of varying frequencies.

Although they address different aspects of the learning process and are therefore complementary to embedding techniques, general online machine learning techniques may also improve convergence of TUL models in an online setting, including replay methods (see e.g. Mnih et al., 2015; Lillicrap et al., 2019; Prabhu et al., 2020) and adaptive optimizers like Hypergradient Descent (Baydin et al., 2018) and DoG (Ivgi et al., 2023).

## 4 Multi-Level Spatial Embedding Sharing

Existing TUL approaches use lookup-table embeddings to encode the spatial information of check-ins. This embedding method uses a matrix $\boldsymbol{Z}$ of shape $|\mathbb{P}| \times d$, where $\mathbb{P}$ is the set of unique POIs and $d$ the embedding dimensionality. The embedding function can be denoted as $z(i) = \boldsymbol{Z}_i$, where $i$ is the index of a POI and $\boldsymbol{Z}_i$ is the $i$-th row vector of the embedding matrix. Given that only a single row is selected for any given location $\boldsymbol{l}$, the sparsity of this approach, defined as the fraction of active parameters in the embedding matrix, is

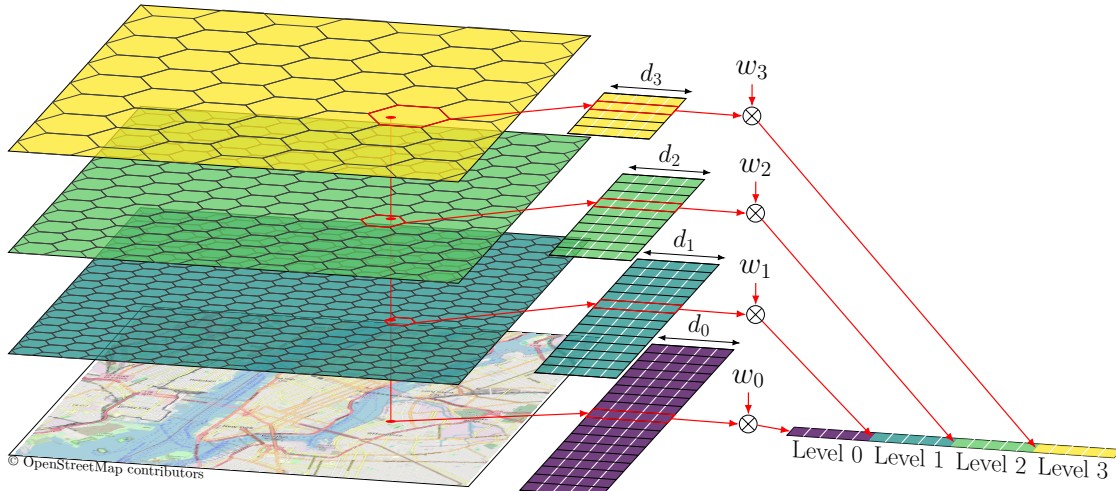

Figure 1: Visualization of the proposed multi-level spatial embedding sharing (MiLES) technique with learnable weights $w_l$ and embedding dimensions $d_l$ for each level $l \in \{0, 1, 2, 3\}$ and $\otimes$ representing scalar multiplication. Embedding dimensions are calculated according to Equation 3 but are drawn equally sized for visual clarity here.

simply $1 - 1/|\mathbb{P}|$. Since mobility data commonly includes thousands of unique POIs, the parameter usage and therefore also the gradients of such embeddings are generally very sparse. Existing methods, such as T3S (Yang et al., 2021) and TULHOR (Alsaeed et al., 2023), attempt to mitigate this issue by sharing embeddings between locations that belong to the same cell in a predefined spatial grid. By grouping multiple locations into shared embedding cells, the number of unique embeddings is reduced, thereby increasing parameter utilization and lowering sparsity to $1 - 1/|\mathbb{H}|$, where $|\mathbb{H}|$ is the number of cells in the grid-based partitioning of the coordinate plane.

This shared embedding approach creates a fundamental trade-off: while grouping locations into cells reduces gradient sparsity, it also reduces the informational content of the embeddings, as multiple distinct locations are now represented by a single vector (see Appendix B.1 for a formal analysis). Embedding techniques that use a fixed level of sharing must therefore balance gradient density with representation specificity. However, in an online learning setting, the importance of each factor likely depends on the stage of the data stream. For instance, after a concept drift occurs, such as the introduction of previously unobserved locations, rapid adaptation becomes crucial, while later on, more detailed, but slower-adapting, features may be preferable.

To address this challenge in online TUL, we propose multi-level spatial embedding sharing (MiLES), which generates embedding features that span a broad range of the density-information spectrum. MiLES is implemented as a drop-in replacement for standard embedding layers, adding minimal inference overhead while reducing the overall parameter count. In the following, we will give an in-depth description of the functionality of MiLES, which is also depicted in Figure 1. Like TULHOR's embedding approach (Alsaeed et al., 2023), MiLES maps locations to a grid created as a tiling of regular hexagons. We use hexagonal grids as they represent Euclidean distances more consistently than square grids (Ke et al., 2019). Unlike TULHOR, MiLES uses multiple mappings with increasing cell sizes and therefore increasing levels of aggregation.

Accordingly, we derive an index $h_l$ for each embedding-level $l \in \{0, 1, ..., l^{(\max)}\}$. This index identifies the specific POI for $l = 0$ and the grid cell containing the check-in location for $l > 0$. For datasets without POI identifiers, such as GeoLife (Zheng et al., 2010), $l = 0$ corresponds to the finest-resolution grid instead.

To accommodate the emergence of previously unobserved locations, the embedding matrices $\boldsymbol{Z}_l$ can be dynamically expanded. To do so, a new randomly initialized row vector is appended to the matrix, whenever a check-in maps to an index $h_l$ that is not currently represented in $\boldsymbol{Z}_l$. This allows the model to adapt to new locations or grid cells as they appear in the stream without requiring a predefined vocabulary size. However, for ease of implementation in our experiments where the maximum grid indices are known a priori,

---

**Algorithm 1:** MiLES: Multi-level Spatial Embedding Sharing

---

**Input:** Check-in $\boldsymbol{c}$, Embedding dims $d_l$
**Parameters:** Embedding matrices $\{\boldsymbol{Z}_l\}$, Level weights $\boldsymbol{w}$
**Output:** Embedding vector $\mathbf{g}$

---

**1** $\boldsymbol{l} \leftarrow \boldsymbol{c}$.coordinates;
**2** $\mathcal{E} \leftarrow [\,]$ ;                                        // Initialize list for level-specific embeddings.
**3 for** $l \leftarrow 0$ **to** $l^{(\max)}$ **do**
**4**     **if** $l = 0$ **and** $c$ *contains POI identifier* $p$ **then**
**5**         $h_l \leftarrow p$;
**6**     **else**
**7**         $h_l \leftarrow \mathrm{GetHexGridIndex}(\boldsymbol{l}, \mathrm{level} = l)$;
**8**     **if** $h_l \geq \mathrm{rows}(\boldsymbol{Z}_l)$ **then**
**9**         $\mathbf{r}_{\mathrm{new}} \leftarrow \mathrm{InitializeVector}(d_l)$;
**10**        $\boldsymbol{Z}_l \leftarrow \mathrm{AppendRow}(\boldsymbol{Z}_l, \mathbf{r}_{\mathrm{new}})$;                    // Expand embedding table.
**11**     $\mathbf{z} \leftarrow \boldsymbol{Z}_l[h_l]$;
**12**     $\mathcal{E} \leftarrow \mathcal{E} \cup \{\mathbf{z} \cdot w_l\}$ ;                        // Apply level weight and append result.
**13** $\mathbf{g} \leftarrow \mathrm{Concat}(\mathcal{E})$;                              // Concatenate results of all levels.
**14 return** $\mathbf{g}$;

---

we pre-allocate $\boldsymbol{Z}_l$ with shape $|\mathbb{H}_l| \times d_l$, where $|\mathbb{H}_l|$ is the number of unique location indices of level $l$. Since only embeddings indexed by the current input receive updates, pre-allocation yields behavior equivalent to dynamic expansion, provided the same parameter initialization is used. In a true deployment scenario where the spatial extent is not known in advance, dynamic expansion would be required, incurring a small per-insertion overhead for appending rows to the embedding matrices but avoiding the memory cost of pre-allocating for unseen regions.

To account for the decreasing informational content with increasing levels of aggregation, we assign smaller dimensions to higher-level embeddings. We compute the individual embedding dimensions as

$$d_l = \left\lfloor \frac{d \cdot \alpha^{-l}}{\sum_{l=0}^{l^{(\max)}} \alpha^{-l}} \right\rfloor, \ \alpha > 1 \tag{3}$$

where $d$ is the dimension of the final embedding and $\alpha$ is a hyperparameter. To reach the total number of dimensions $d$ we add the remaining dimensions to the initial embedding level. Based on the results of a hyperparameter search (see Table 11), we use $\alpha = 2$ in our experiments.

For the final embedding, we concatenate all level-specific embeddings, each weighted by a learnable parameter $w_l$. Using $\|$ to represent vector concatenation, we define the embedding function $g$ as

$$g(\boldsymbol{h}; \boldsymbol{Z}_0, \boldsymbol{Z}_1, ..., \boldsymbol{Z}_{l^{(\max)}}, \boldsymbol{w}) = \overset{l^{(\max)}}{\underset{l=0}{\Big\|}} \boldsymbol{Z}_{l,h_l} \cdot w_l, \tag{4}$$

where $\boldsymbol{Z}_{l,h_l}$ is the row vector of $\boldsymbol{Z}_l$ located at index $h_l$.

We select concatenation instead of summation to aggregate the level-specific embeddings to avoid interference between levels. For a fixed total embedding dimension, this approach also significantly reduces the number of learnable parameters, since higher-level embeddings are shared across many locations.

By multiplying the level-specific embeddings $\boldsymbol{Z}_{l,h_l}$ with learnable parameters $w_l$, we allow the average magnitude of the embedding vectors to be optimized on a per-level basis throughout the online learning process. These global scalar weights are updated via gradient descent alongside all other model parameters, allowing the model to learn the relative importance of each spatial level. Because the weights evolve over the course

of training, they can implicitly reflect shifts in which spatial granularity is most informative as more data is observed.

Algorithm 1 summarizes MiLES' embedding construction for a single check-in. To embed a full trajectory $T = (\boldsymbol{c}_1, \ldots, \boldsymbol{c}_n)$, this process is applied independently to each check-in: the location $\boldsymbol{l}_i$ is passed through MiLES to obtain a spatial embedding $\boldsymbol{g}_i$, while the timestamp $t_i$ is encoded via an hour-of-day lookup embedding. These are concatenated to form the check-in representation, which serves as a drop-in replacement for the spatial input of any sequence-based TUL backbone. The sequence of check-in representations is then processed by the backbone (e.g., BiTULER's bidirectional LSTM) to produce a trajectory-level representation for user classification. All parameters including MiLES's embedding matrices $\boldsymbol{Z}_l$, level weights $\boldsymbol{w}$, temporal embeddings, and backbone parameters are trained jointly via the loss in Equation (2).

## 5 Experiments

To evaluate the impact of the proposed MiLES approach and its individual components on the performance of existing TUL models in a data stream setting, we perform a series of experiments.

We use the widely adopted Foursquare-NYC, Foursquare-TKY (Yang et al., 2015) and GeoLife (Zheng et al., 2010) datasets. Following standard preprocessing (Chen et al., 2022), we split each trajectory into shorter segments with a maximum length of 24 hours for Foursquare-NYC and Foursquare-TKY and 3 hours for GeoLife, and selecting the most active users. For GeoLife, we additionally subsample check-ins at one-minute intervals. For more information on the selected datasets, see Table 1.

For optimization, we use Adam (Kingma & Ba, 2017). Across datasets, the input modalities include GPS coordinates, timestamps, and POI identifiers, except in GeoLife, which lacks POI information. We also provide hour-specific lookup embeddings, following prior work (see, e.g. Chen et al., 2022; Miao et al., 2020). We do not incorporate components that rely on data beyond the trajectories themselves (e.g., TULHOR's mobility flows), as such information is absent from standard trajectory datasets and lies outside the scope of our embedding-based approach.

We tune all hyperparameters by maximizing mean prequential top-1 accuracy on the first 5,000 trajectories from the 400-user Foursquare-TKY stream. This procedure is applied consistently for all models and methods including MiLES, with tuning data excluded from evaluation. We deliberately restrict tuning to a small initial segment from a single dataset to reflect the constraints of a realistic online deployment, where labeled data for hyperparameter optimization from the same data source is often times unavailable at the outset. Using these same hyperparameters for all datasets without adjusting them means that no method is optimized individually for each experimental condition, and any disadvantage from cross-dataset transfer affects all configurations equally. The comparison of online and batch learning scenarios in Section 6.1 uses a separately tuned hyperparameter configuration, under which MiLES's gains remain comparable in magnitude (see Table 8) on all datasets including GeoLife. This suggests that the results are not sensitive to the specific hyperparameter setting and that the shared protocol does not disproportionately affect any method.

For methods requiring historical data (MainTUL and DeepTUL), we maintain a buffer of the last 1,000 trajectories. To determine embedding sharing levels, we partition the map into multiple hexagonal grids and tune the number of rows using the same protocol. The best configuration uses three levels with 200 rows at the base, halving the number of rows at each subsequent level. For GeoLife, POI embeddings are replaced with a grid-based embedding of 800 rows.

Table 1: Datasets used for experimental evaluation.

| Dataset | Foursquare-NYC | | Foursquare-TKY | | GeoLife | |
|---|---|---|---|---|---|---|
| Users | 800 | 400 | 800 | 400 | 150 | 75 |
| Trajectories | 61,218 | 35,510 | 70,007 | 44,955 | 25,611 | 23,290 |
| Check-Ins | 196,435 | 137,886 | 324,564 | 248,771 | 1,284,208 | 1,187,510 |
| POIs | 34,383 | 25,443 | 38,212 | 28,286 | — | — |

A complete list of hyperparameters is provided in Table 11. To ensure fairness, the total embedding dimensionality is fixed across methods (after baseline tuning), so that performance differences reflect only the embedding strategy rather than representational capacity.

To address the impact of MiLES' hyperparameters, we conduct a detailed analysis of the effect of the number of embedding levels and grid resolutions on model performance, as shown in Figure 5. Standard deviations are omitted from tables for brevity. Unless stated otherwise, the reported results represent averages over 5 independent runs

## 6  Results

In this section, we present a comprehensive evaluation of MiLES to validate its effectiveness and robustness in online learning scenarios. Our analysis is structured as follows: First, we demonstrate the practical benefits of MiLES by integrating it into several state-of-the-art TUL models. Following this, we compare its performance against alternative embedding strategies and general online learning techniques. Finally, we conduct an in-depth analysis of the MiLES architecture itself, using a detailed ablation study and statistical tests to isolate and verify the contribution of each of its individual components.

**Enhancing TUL Models with MiLES**  We evaluated various TUL approaches with their original location embedding techniques for online learning applications. The results of these experiments for the higher dataset variants with higher user counts are depicted in the upper section of Table 2 (see Table 12 for all results). As Table 2 shows, the bidirectional-LSTM-based BiTULER achieved the overall best performance on all datasets, except for GeoLife where DeepTUL yielded a higher top-1 accuracy and macro F1 score. This exception likely stems from BiTULER being the most lightweight model with the fewest parameters, enabling faster adaptation. Similarly, the generally lower performance of the transformer-based models (MainTUL, T3S, and TULHOR) compared to their RNN-based counterparts may reflect the challenges of adapting more complex architectures in online learning scenarios. We then replaced the original embedding modules in each TUL model with MiLES, keeping the total embedding dimension constant. The bottom section of Table 2 shows the relative change in performance metrics measured in percentage points achieved by using MiLES.

The most substantial improvements were on GeoLife, where MiLES increased top-1 accuracy by up to 8.72 percentage points (a 23% relative gain) and top-5 by up to 6.98 points. This larger gain for GeoLife likely stems from its lack of POI information, where MiLES' additional embedding levels provide particular value compared to existing techniques' original embedding approach.

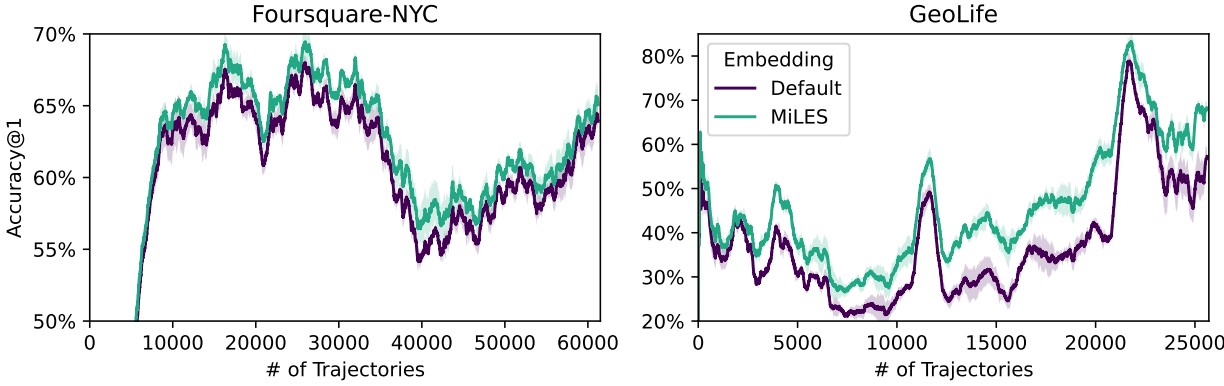

Figure 2: Rolling top-1 accuracy for BiTULER with and without MiLES on Foursquare-NYC (window size 5,000) and GeoLife (window size 1,000). Shaded areas indicate the $4\sigma$ interval over all random seeds. The increase in accuracy toward the end of the GeoLife stream coincides with a reduction in diversity of user labels within the evaluation window, which lowers the effective difficulty of the classification task in this segment (see Figure 11).

Table 2: Top-1 accuracy, top-5 accuracy and macro F1 score [%] averaged over prequential evaluation runs for TUL models with their original embedding technique (Default) and performance gains when using our technique instead ($\Delta$ MiLES).

| Dataset | Foursquare-NYC | | | Foursquare-TKY | | | GeoLife | | |
|---|---|---|---|---|---|---|---|---|---|
| Model | Acc@1 | Acc@5 | F1 | Acc@1 | Acc@5 | F1 | Acc@1 | Acc@5 | F1 |
| Default | | | | | | | | | |
| BiTULER | **60.12** | **67.20** | **57.83** | **63.16** | **74.91** | **61.06** | **37.56** | 70.85 | 26.69 |
| TULVAE | 59.79 | 66.77 | 57.32 | 55.82 | 66.23 | 51.68 | 37.08 | 70.45 | 25.25 |
| DeepTUL | 58.72 | 65.48 | 56.60 | 61.17 | 72.49 | 59.10 | 36.32 | **72.64** | **29.82** |
| MainTUL | 55.67 | 62.61 | 53.01 | 59.53 | 71.89 | 57.09 | 34.00 | 70.26 | 21.76 |
| T3S | 52.98 | 60.28 | 49.50 | 56.41 | 69.26 | 53.24 | 35.25 | 71.11 | 21.52 |
| TULHOR | 53.85 | 61.13 | 50.40 | 56.61 | 69.61 | 53.45 | 34.65 | 72.46 | 24.92 |
| $\Delta$ MiLES | | | | | | | | | |
| BiTULER | +1.49 | +3.58 | +1.71 | +1.40 | +2.85 | +1.30 | +8.72 | +6.98 | +6.26 |
| TULVAE | +1.79 | +3.73 | +2.04 | +3.07 | +4.43 | +3.34 | +8.01 | +6.57 | +4.74 |
| DeepTUL | +1.06 | +3.04 | +1.26 | +0.84 | +2.37 | +0.77 | +8.59 | +6.31 | +5.97 |
| MainTUL | +1.44 | +4.33 | +1.52 | +1.62 | +3.45 | +1.45 | +8.19 | +6.18 | +5.50 |
| T3S | +1.71 | +3.13 | +2.10 | +1.70 | +2.79 | +1.91 | +6.98 | +4.80 | +5.18 |
| TULHOR | +1.71 | +3.16 | +2.11 | +1.85 | +2.80 | +2.03 | +8.20 | +5.03 | +5.68 |

Notably, improvements were consistent across all models, including T3S and TULHOR, which already incorporated single-level grid-based embeddings, demonstrating the significant benefits of MiLES' multi-level approach in online learning settings. Furthermore, since the embedding dimensionality was tuned only for the default embedding approach, the reported gains are conservative and could likely be improved with MiLES-specific tuning.

In addition to these aggregate results, Figure 2 shows the evolution of rolling top-1 accuracy over the data stream for BiTULER with and without MiLES on Foursquare-NYC and GeoLife. On Foursquare-NYC, MiLES maintains a relatively consistent advantage throughout the stream rather than concentrating its gains in any particular segment, suggesting that the benefits of multi-level sharing persist as the model accumulates training signal. The gap appears to widen slightly during the accuracy dip around trajectory 40,000, which coincides with elevated distributional shift in POI visits (see Figure 10), suggesting that MiLES may be particularly beneficial during periods of non-stationarity. On GeoLife, the advantage of MiLES grows over the course of the stream, consistent with the shared grid-level embeddings accumulating increasingly informative spatial representations as more locations are observed.

To evaluate the impact of MiLES on model behavior after a concept drift, we removed all check-ins in the eastern half of the Foursquare-NYC dataset from the first 30,000 trajectories, so that the model would suddenly encounter a significant fraction of previously unobserved POIs. This emulates an abrupt concept drift that may occur, for instance, if a location-based service expands to cover a larger area.

Table 3: Novelty introduced by the simulated concept drift on Foursquare-NYC at each MiLES embedding level. *New locations* are indices (POIs or grid cells) encountered after the drift that were absent from the pre-drift stream. The rightmost column restricts this count to the first 1,000 post-drift trajectories.

| Level | Total Locations | New Locations (Total) | New Locations (First 1k Tr.) |
|---|---|---|---|
| 0 | 28,214 | 12,321 | 745 |
| 1 | 7,549 | 3,743 | 360 |
| 2 | 4,129 | 2,012 | 269 |
| 3 | 1,716 | 804 | 177 |

Table 4: Post-drift performance for BiTULER with and without MiLES on the simulated Foursquare-NYC drift experiment. Rolling accuracy is computed with a window of 1,000 trajectories. Recovery denotes the number of post-drift trajectories until rolling accuracy returns to 95% of the pre-drift value.

|  | Post-drift minimum [%] | | Recovery [trajectories] | |
|  | Acc@1 | Acc@5 | Acc@1 | Acc@5 |
|---|---|---|---|---|
| Default | 36.68 | 42.68 | 1,557 | 1,607 |
| MiLES | 39.44 | 46.90 | 1,459 | 1,640 |

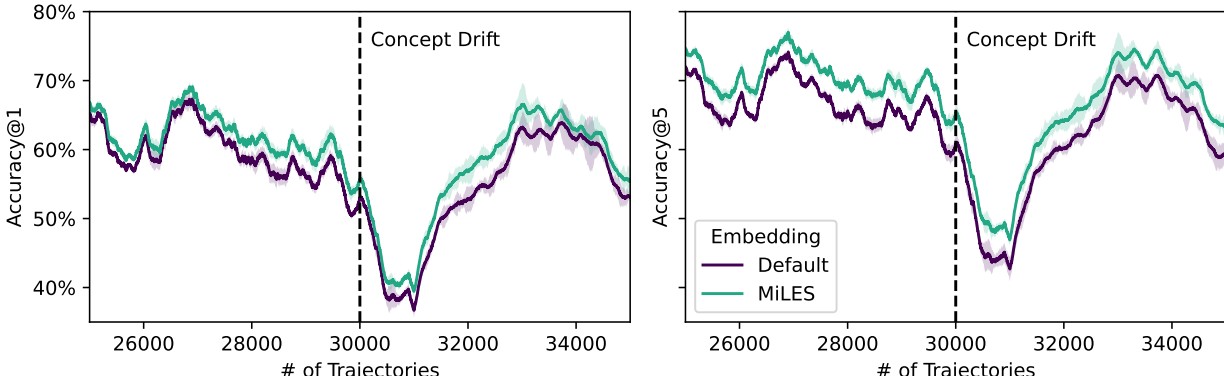

Figure 3: Impact of simulated spatial expansion on Foursquare-NYC. After 30000 trajectories, the model was suddenly exposed to check-ins in the eastern half of the city, which were excluded from the initial trajectories. Both metrics were calculated with a sliding window of size 1000. The highlighted areas mark the $2\sigma$ interval over all random seeds.

Table 3 quantifies the novelty introduced by this intervention at each MiLES embedding level. At the POI level, over a third of check-ins in the first 1,000 post-drift trajectories reference entirely new locations, while at the coarsest grid level this fraction drops below a quarter. This illustrates how MiLES could mitigate the impact of novel locations. While POI-level embeddings must be learned from scratch, coarser levels already carry informative spatial representations from nearby, previously observed check-ins. The performance impact is shown in Figure 3 and quantified in Table 4. MiLES maintains a higher performance floor after the drift across both metrics, while recovery times are comparable between configurations. This is consistent with the novelty analysis: the coarser grid levels provide a more informed starting point for newly introduced locations, raising the minimum but not fundamentally accelerating learning at the POI level.

Finally, we analyzed the computational efficiency of MiLES using the BiTULER model. As detailed in Table 5, MiLES significantly reduces the model size, decreasing the number of embedding parameters by approximately 38% on the Foursquare datasets (e.g., from 35.2M to 22.0M on NYC). Furthermore, the lower parameter counts translate to faster model updates and therefore shorter runtimes per online learning iteration, due to the more efficient updates outweighing MiLES' small inference overhead. Since the parameter

Table 5: Number of embedding parameters and wall time per iteration for BiTULER using MiLES or its original embedding technique. Measured on a system with an Intel i5-9600K CPU and an Nvidia RTX 3090 GPU.

| Dataset | Foursquare-NYC | | Foursquare-TKY | | GeoLife | |
| Embedding | # Parameters | Wall Time | # Parameters | Wall Time | # Parameters | Wall Time |
|---|---|---|---|---|---|---|
| Default | 35.21M | 7.99 ms | 39.13M | 8.50 ms | 4.74M | **11.27 ms** |
| MiLES | **21.97M** | **7.38 ms** | **24.32M** | **7.81 ms** | **3.28M** | 11.66 ms |

count of the default embeddings on Geolife is already relatively low, we observe a slight increase in wall time for this dataset. This increase could potentially be mitigated with an optimized implementation of MiLES.

**Comparison with common online learning techniques**  We further evaluated our approach against alternative embedding-, experience replay- and adaptive optimization methods. The results, averaged across all models from Table 2, are shown in Table 6. Among the embedding techniques, MiLES achieved the best overall performance across all datasets and metrics. It consistently outperformed the baseline POI-based lookup embeddings, as well as the hybrid linear and Fourier embeddings. These two alternatives combined POI-based embeddings with either a learnable linear projection or a Fourier feature encoding (Tancik et al., 2020) of the location coordinates, replacing the higher-level embeddings used in MiLES. Notably, both hybrid approaches underperformed compared to the simpler POI-only embeddings, suggesting that basic coordinate projections are less effective than fully dedicating the embedding space to POI-specific representations. For memory replay strategies, we compared a FIFO buffer and a random class-balanced buffer, each limited to 1,000 past trajectories, with one sample replayed per training step. Both replay strategies improved performance over the default configuration with the models original embeddings and without replay. When combining MiLES with FIFO replay (FIFO + MiLES), we observed further gains across all datasets and metrics, demonstrating that MiLES remains effective when paired with replay techniques. The adaptive optimizers DoG and AdamHD performed significantly worse than the standard Adam optimizer, used for all other methods. We attribute this to the high gradient variance introduced by training on individual samples instead of mini-batches, which destabilizes the learning rate adaptation in both methods.

Table 6: Top-1 accuracy, top-5 accuracy and macro F1 score [%] of different embedding-, experience replay- and adaptive optimization methods, averaged across all models shown in Table 2. See Table 13 for the full results. The default configuration uses POI-based lookup embeddings and the Adam optimizer, without replay. Individual methods replace the default components. Linear and Fourier embeddings combine POI lookups with a linear or Fourier projection of coordinates, each contributing half the embedding dimensions.

| Dataset | Foursquare-NYC | | | Foursquare-TKY | | | GeoLife | | |
|---|---|---|---|---|---|---|---|---|---|
| Method | Acc@1 | Acc@5 | F1 | Acc@1 | Acc@5 | F1 | Acc@1 | Acc@5 | F1 |
| Default | 56.85 | 63.91 | 54.11 | 58.78 | 70.73 | 55.94 | 35.81 | 71.30 | 25.00 |
| Embedding | | | | | | | | | |
| Linear | 50.74 | 59.21 | 47.51 | 53.48 | 66.25 | 50.16 | 35.02 | 70.62 | 22.66 |
| Fourier | 45.31 | 57.79 | 41.68 | 51.18 | 67.72 | 47.49 | 34.54 | 70.56 | 23.26 |
| MiLES | **58.39** | **67.41** | **55.90** | **58.33** | **71.68** | **55.40** | **43.93** | **77.27** | **30.55** |
| Replay | | | | | | | | | |
| FIFO | 57.38 | 64.53 | 55.24 | 57.66 | 69.43 | 55.04 | 37.33 | 70.87 | 25.82 |
| Balanced | 57.19 | 64.26 | 54.72 | 56.96 | 68.66 | 53.95 | 36.40 | 68.37 | 25.36 |
| FIFO+MiLES | **58.94** | **67.88** | **56.87** | **60.53** | **73.85** | **57.74** | **45.86** | **77.49** | **31.87** |
| Optimizer | | | | | | | | | |
| DoG | 36.79 | 42.57 | 33.82 | 35.95 | 45.74 | 34.00 | 29.12 | 60.59 | 21.14 |
| AdamHD | 45.94 | 51.80 | 43.18 | 40.15 | 49.03 | 37.89 | 22.24 | 48.18 | 17.09 |

**Analysis of MiLES Components**  To assess the contribution of individual components in MiLES, we conducted ablation studies by systematically removing embedding levels (-L1, -L2, -L3), the learnable weighting parameters $w_l$ (-WL), and the level-dependent embedding dimensions (-VD). When excluding an embedding level, we maintained the total embedding dimension and adjusted the dimensions of the remaining levels by omitting the affected level from Equation 3. In the -VD setting, we distributed the total embedding size equally among levels.

Figure 4 plots the macro F1 scores and top-5 accuracy of our full approach and each ablation, revealing that the complete embedding is consistently on or near the Pareto frontier and achieves a favorable balance

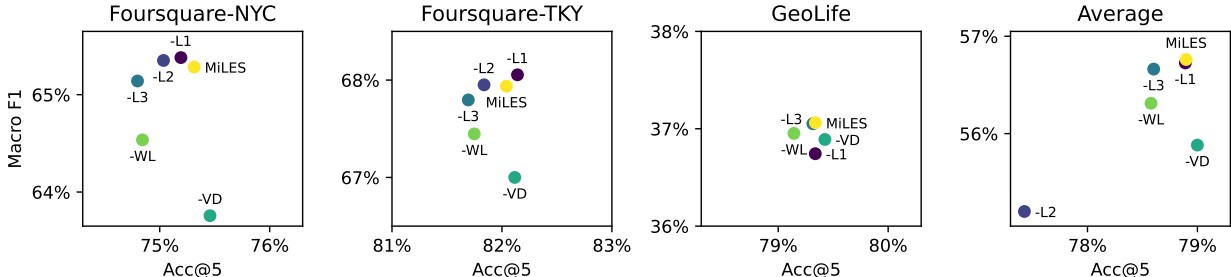

Figure 4: Macro F1 score and top-5 accuracy results of ablation study for MiLES with BiTULER as the underlying model. Metrics are averaged over both variants of each dataset. -L2 was omitted in the GeoLife and average results for visual reasons, since the removal of the level two embeddings resulted in a large decrease in top-5 accuracy on GeoLife.

between the two metrics. This is particularly evident in the average performance across all datasets, where removing any component results in a noticeable drop in at least one metric. For instance, removing the variable dimensionality (-VD) leads to a large decrease in macro F1 for only a marginal gain in top-5 accuracy. Only the removal of the highest resolution grid embeddings (-L1) results in an insignificant drop in average performance metrics. The dataset-specific plots show that the highest-resolution grid (-L0) offers no additional benefit on Foursquare-TKY, likely due to the datasets high density of check-ins per POI. In all other scenarios, however, each component proves its value.

To validate these observations statistically, we performed one-sided Wilcoxon signed-rank tests (Table 7). For each ablation, dataset and metric, we constructed paired observations by matching the full MiLES configuration against the ablated configuration for the same combination of dataset variant and seed. This yielded 20 pairs per test (two dataset variants × 10 random seeds). The one-sided test evaluates whether the paired differences (full MiLES minus ablated) are systematically positive against the null hypothesis that these differences are distributed symmetrically around zero.

On the POI-rich Foursquare datasets, the additional embedding levels primarily yield highly significant gains in top-5 accuracy ($p < 0.001$). Conversely, on GeoLife, which lacks POI data, these levels provide statistically significant improvements to top-1 accuracy and macro F1 score. Similarly, the variable dimensionality (-VD) and learnable weights (-WL) offer highly significant contributions to performance across most datasets and metrics.

In conclusion, while the performance gain from any single component may seem moderate in isolation, our analysis shows that all components contribute significantly to the model's overall robustness. It is also

Table 7: P-values of one-sided Wilcoxon signed-rank tests, with the null hypothesis being that removing the respective component does not degrade the respective performance metric. Each paired observation is the metric difference (full MiLES minus ablated) from a single prequential evaluation run, identified by a unique combination of dataset variant and seed. With 2 variants and 10 seeds per dataset, each test is based on N = 20 paired observations. Significant values are highlighted, with higher significance levels receiving a darker hue.

| Dataset | Foursquare-NYC | | | Foursquare-TKY | | | GeoLife | | |
|---|---|---|---|---|---|---|---|---|---|
| Method | Acc@1 | Acc@5 | F1 | Acc@1 | Acc@5 | F1 | Acc@1 | Acc@5 | F1 |
| -L1 | 0.996 | <0.001 | 0.992 | 0.998 | 0.999 | 0.996 | 0.022 | 0.712 | 0.003 |
| -L2 | 0.938 | <0.001 | 0.973 | 0.411 | <0.001 | 0.727 | <0.001 | <0.001 | <0.001 |
| -L3 | <0.001 | <0.001 | <0.001 | <0.001 | <0.001 | <0.001 | <0.001 | 0.273 | 0.364 |
| -VD | <0.001 | 0.969 | <0.001 | <0.001 | 0.999 | <0.001 | 0.152 | 0.905 | 0.101 |
| -WL | <0.001 | <0.001 | <0.001 | <0.001 | <0.001 | <0.001 | <0.001 | 0.001 | 0.147 |

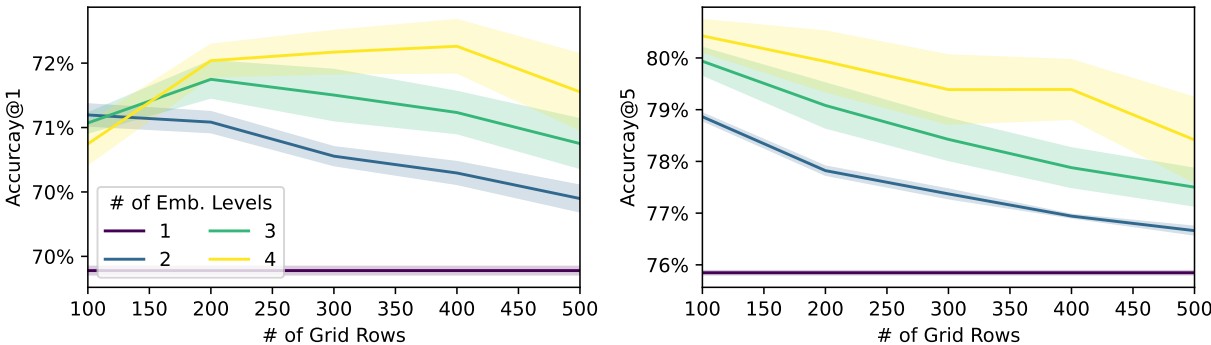

Figure 5: Performance on the 400 user Foursquare-NYC dataset, relative to number of embedding levels and rows in the base-level grid using BiTULER and MiLES with fixed level weights $w_l$. Shaded areas represent the $1\sigma$ range.

worth noting that these gains are conservative. For the sake of a fair comparison, the total embedding dimension was kept constant, which reduced the capacity of the powerful POI-based embeddings when adding new levels. We hypothesize that tuning the dimensionality specifically for MiLES would widen the performance gap further. Furthermore, even a component like L1, whose average gain is small, is justified by its negligible computational cost of a single table lookup, as it provides robustness on datasets like GeoLife without sacrificing efficiency.

In another experiment, we evaluated MiLES for varying embedding aggregation levels and varying grid resolutions with the level-specific weights $w_l$ fixed at 1. For this, we used the Foursquare-NYC dataset with 400 users and BiTULER as the underlying classifier. Following previous experiments, level two used grids at half the base resolution and level three at quarter resolution. The results in Figure 5 indicate that increasing the number of embedding levels improves performance, provided that the base-level grid has a sufficiently high resolution.

In particular, the four-level embedding module achieves the highest top-1 and top-5 accuracy when the base-level grid contains at least 200 rows. This suggests that MiLES is robust to variations in base grid resolution and that all embedding levels contribute meaningfully to performance. The effect of base grid resolution differs between top-1 and top-5 accuracy. While top-1 accuracy peaks at 400 grid rows for the four-level configuration, top-5 accuracy decreases steadily as resolution increases. This is consistent with earlier observations that coarser sharing benefits group-level prediction and therefore top-5 accuracy.

To analyze the dynamics of the learnable level-weighting parameters $w_l$, we tracked their values across multiple prequential evaluation runs on the 400-user Foursquare-NYC dataset. Figure 6 shows the evolution

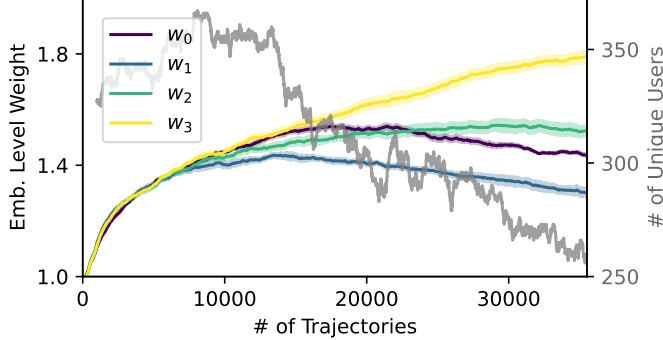

Figure 6: Mean weighting factors $w_0, ..., w_3$ and number of unique users within the last 1,000 trajectories for Foursquare-NYC. Shaded areas represent the $1\sigma$ range.

of these weights alongside the number of unique users in the 1,000 most recent trajectories.

On average, all level-specific weights increase throughout training, effectively increasing the learning rate of the embedding parameters, as explained in Section 4. However, there are differences in how individual weights evolve. Initially, the weights associated with lower sparsity embedding levels ($w_2$ and $w_3$) increase slightly faster than the POI-based embedding weight ($w_0$). After 5,000 trajectories, $w_0$ surpasses $w_1$ and $w_2$, suggesting that more specific embeddings become beneficial at this stage of online learning. However, after about 15,000 samples, both $w_0$ and $w_1$ begin to decline. This likely corresponds to an earlier concept drift, characterized by a reduction in unique users, making the coarser embeddings ($w_2$ and $w_3$) sufficient for user selection.

**MiLES in Batch Learning**  To disentangle the contribution of MiLES's representational capacity from its interaction with the online learning setting, we compare the performance gains of MiLES over the baseline in both batch and online evaluation modes. Hyperparameters for this comparison were tuned separately from the main online experiments: we split the first 5,000 trajectories from Foursquare-TKY into training and test sets and selected the values providing the best accuracy on the test set after 20 training epochs. These batch-tuned hyperparameters were then used for both the batch and online evaluation modes.

As shown in table 8, MiLES improves performance in both settings, confirming that the multi-level architecture provides a general representational benefit that is not exclusive to online learning. However, the gains are consistently larger in the online setting, particularly for top-5 accuracy and macro F1. This discrepancy is most pronounced on GeoLife, where MiLES yields a top-1 accuracy improvement of +22.10% in the online setting compared to +4.81% in batch learning. We attribute this to the greater severity of gradient sparsity in the online regime. Since each trajectory is observed only once, POI embeddings receive far fewer gradient updates than in batch training, where multiple epochs ensure that even infrequent locations are eventually well-represented. The effect is amplified on GeoLife, where trajectories are less spatially diverse, meaning that a larger fraction of POIs remain underrepresented at any given point in the stream. In this regime, the denser gradient signal provided by MiLES's shared grid-based levels becomes especially valuable.

Additionally, MiLES's learnable level weights (Figure 6) allow the model to dynamically adjust the relative contribution of each embedding level throughout the stream. This property is inherently more relevant in a non-stationary online setting than in batch learning, where the weights converge to fixed values after training. This experiment also provides additional evidence for the robustness of MiLES to hyperparameter choices. The batch-tuned hyperparameters differ substantially from the online-tuned ones used in the main experiments, yet MiLES yields consistent improvements under both configurations across all three datasets.

Table 8: Relative performance gain [%] of MiLES over the baseline in batch and online evaluation. We report relative rather than absolute gains to facilitate comparison across the two settings, which differ in baseline performance levels. Hyperparameters were tuned for the batch setting and used for both modes.

| Dataset | Foursquare-NYC | | | Foursquare-TKY | | | GeoLife | | |
| Scenario | Acc@1 | Acc@5 | Macro F1 | Acc@1 | Acc@5 | Macro F1 | Acc@1 | Acc@5 | Macro F1 |
|---|---|---|---|---|---|---|---|---|---|
| Batch | +4.24 | +6.41 | +3.84 | +3.97 | +4.80 | +3.93 | +4.81 | +2.52 | +8.10 |
| Online | +4.32 | +8.04 | +5.31 | +4.44 | +6.62 | +4.69 | +22.10 | +10.03 | +21.57 |

## 7 Generalization to Destination Prediction

To demonstrate the general applicability of MiLES beyond Trajectory-User Linking, we conducted additional experiments on the task of destination prediction, which has significant practical importance for applications such as personal navigation systems or ride sharing platforms (Endo et al., 2017). The objective of this task is to predict the coordinates of the final check-in $c_n$, of a trajectory given an initial sub-trajectory of check-ins $[c_0, c_1, \ldots, c_k]$, where the lengths of the complete trajectory $n$ and the input sequence $k$ may vary.

We emphasize that we use destination prediction solely to assess the generality of MiLES, not to establish new state-of-the-art results for this task.

## 7.1 Experimental Setup

Our experimental setup is based on the ECML/PKDD 2015 discovery challenge on taxi destination prediction (de Brébisson et al., 2015). For each dataset, we generated a stream of 50,000 pairs of partial trajectories and their final destinations by sampling sub-trajectories at random points. For the GeoLife dataset, we first filtered out check-ins outside the greater Beijing area and subsampled the remaining data at 10-minute intervals.

We framed the task as a regression problem. To ensure that the Euclidean distance accurately reflect real-world distances, we projected the destination coordinates $l_n = [\text{longitude}_n, \text{latitude}_n]$ into a local planar coordinate system. All models were then trained to predict these projected coordinates, allowing the prediction error to be calculated as the straight-line distance.

In terms of model architecture, we also employed a similar approach to that used in de Brébisson et al. (2015): We encode partial trajectories, as well as the weekday and user-ID associated with the respective trajectory. For the latter, we use embedding vectors with 256 dimensions each. All encoded features are then fed to a fully connected layer, producing the final prediction.

Previous work used a variety of neural architectures like multi-layer perceptrons (MLPs) (de Brébisson et al., 2015), convolutional neural networks (CNNs) (Lv et al., 2018) or LSTMs (Endo et al., 2017; Ebel et al., 2020) for the purpose of destination prediction. Based on this, we used either a bidirectional LSTM (BiLSTM), a convolutional neural network (CNN), a transformer (Transformer) or an MLP to encode the partial trajectories For the latter, we computed the average of all check-in embeddings to aggregate the partial trajectory.

We tune the hyperparameters of all models and techniques using the initial 5000 partial trajectories of the 400 user Foursquare-TKY dataset.

## 7.2 Results

We evaluated all models using either a default single-level POI lookup embedding or the proposed MiLES embedding. Table 9 presents the mean, median (P50), and 90th percentile (P90) of the prediction errors, averaged over five prequential evaluation runs.

Among the baseline models, the CNN performed best on the Foursquare datasets, while the BiLSTM was superior on GeoLife. The Transformer model yielded the worst results across all datasets, which is likely

Table 9: Mean, 50th-, and 90th percentile of distances between predicted and actual destinations in kilometers for models using a basic lookup embedding (Default) and performance gains when using our technique ($\Delta$ MiLES).

| Dataset | Foursquare-NYC | | | Foursquare-TKY | | | GeoLife | | |
|---|---|---|---|---|---|---|---|---|---|
| Model | Mean | P50 | P90 | Mean | P50 | P90 | Mean | P50 | P90 |
| | | | | Default | | | | | |
| BiLSTM | 7.21 | 5.37 | 20.03 | 6.97 | 5.60 | 17.68 | **5.96** | **4.29** | **16.90** |
| CNN | **6.51** | **4.81** | **18.28** | **6.57** | **5.21** | **16.86** | 6.33 | 4.56 | 17.87 |
| MLP | 7.80 | 6.14 | 20.08 | 7.53 | 6.23 | 18.09 | 7.12 | 5.39 | 19.10 |
| Transformer | 8.90 | 7.26 | 21.81 | 8.11 | 6.93 | 18.58 | 7.54 | 5.81 | 19.97 |
| | | | | $\Delta$ MiLES | | | | | |
| BiLSTM | -0.72 | -0.65 | -1.56 | -0.24 | -0.29 | -0.24 | -0.09 | -0.07 | -0.34 |
| CNN | -0.13 | -0.18 | -0.11 | 0.05 | 0.04 | 0.13 | -0.20 | -0.20 | -0.28 |
| MLP | -0.55 | -0.52 | -0.99 | -0.16 | -0.18 | -0.21 | -0.16 | -0.17 | -0.25 |
| Transformer | -1.01 | -0.97 | -1.89 | -0.48 | -0.51 | -0.59 | -0.09 | -0.09 | -0.17 |

Table 10: Mean, 50th-, and 90th percentile of distances between predicted and actual destinations in kilometers for different embedding- and experience replay methods, averaged across all models shown in Table 9. [†]The default configuration uses POI-based lookup embeddings without replay.

| Dataset | Foursquare-NYC | | | Foursquare-TKY | | | GeoLife | | |
|---|---|---|---|---|---|---|---|---|---|
| Method | Mean | P50 | P90 | Mean | P50 | P90 | Mean | P50 | P90 |
| Default[†] | 7.61 | 5.90 | 20.05 | 7.47 | 6.14 | 18.22 | 6.74 | 5.01 | 18.46 |
| Embedding | | | | | | | | | |
| Linear | 7.13 | 5.56 | **18.63** | 7.20 | 5.92 | **17.55** | 6.83 | 5.12 | 18.53 |
| Fourier | **7.00** | 5.34 | 18.88 | 7.14 | 5.80 | 17.73 | 6.69 | 4.96 | 18.41 |
| MiLES | **7.00** | **5.32** | 18.91 | **7.09** | **5.76** | 17.57 | **6.60** | **4.88** | **18.20** |
| Replay | | | | | | | | | |
| FIFO | 7.39 | 5.52 | 20.35 | 7.30 | 5.82 | 18.64 | 6.37 | 4.50 | 18.33 |
| Random | 7.39 | 5.50 | 20.41 | 7.31 | 5.82 | 18.62 | 6.31 | 4.43 | 18.27 |
| FIFO+MiLES | **6.82** | **4.97** | **19.35** | **7.10** | **5.56** | **18.45** | **6.09** | **4.26** | **17.67** |

attributable to the short average length of the partial trajectories (e.g., only five check-ins for Foursquare-NYC), limiting the effectiveness of its self-attention mechanism.

Consistent with our TUL experiments, integrating MiLES improved destination prediction performance across most models, metrics, and datasets. The Foursquare datasets, which have fewer check-ins per trajectory, benefited most, with MiLES achieving reductions in mean error of up to 11%. The performance gains were less pronounced for the CNN model. On Foursquare-TKY, the CNNs performance even slightly declined.

We hypothesize that this is because the local connectivity of convolutional kernels already causes some similarity between embeddings of nearby POIs, an effect that partially overlaps with the explicit spatial sharing in MiLES.

We further compared MiLES against alternative embedding and experience replay techniques, with results averaged over all models reported in Table 10. While linear embeddings achieved a lower 90th percentile error on the Foursquare datasets, MiLES consistently demonstrated the best performance in terms of mean and median prediction error compared to the default lookup and other hybrid embeddings.

Experience replay strategies significantly reduced mean and median errors. However, they did not improve the 90th percentile error, suggesting that while replay reinforces common travel patterns, it may not help with predicting less frequent or novel destinations that constitute the long-tail of the error distribution.

Notably, the combination of MiLES with a FIFO replay buffer (FIFO + MiLES) achieved the best results across nearly all metrics and datasets, once again demonstrating that MiLES can be used to complement and enhance other online learning strategies.

## 8 Conclusion

In this paper, we investigated online learning for Trajectory-User Linking on streaming trajectory data. To the best of our knowledge, this work presents the first systematic study of TUL in an online learning setting. We introduced Multi-Level Spatial Embedding Sharing (MiLES), an embedding method that balances representational density and specificity through a weighted, multi-scale architecture, with particular benefits in online learning. Our experimental evaluation of the approach provided two key findings. First, in the context of TUL, MiLES consistently enhanced the performance and efficiency of several state-of-the-art models. Second, we demonstrated the generalizability of our method on the task of destination prediction, where MiLES again improved prediction accuracy across multiple backbone architectures. Taken together, these findings establish MiLES as a broadly applicable embedding technique for data-scarce mobility set-

tings. Our controlled drift experiment shows that MiLES maintains higher performance when previously unobserved locations enter the stream. The rolling accuracy analysis confirms that this advantage persists under naturally occurring distributional shifts in unmodified datasets. However, the extent to which this benefit applies to different types and magnitudes of drift is a subject for future work.

## 9 Limitations

Although MiLES shows significant improvements in performance and efficiency for trajectory-user linking and destination prediction, its core inductive bias that spatially proximate locations share useful structure may not apply equally in all settings. For example, when neighboring locations serve fundamentally different functional roles despite their proximity (e.g., a transport hub adjacent to a residential area), the shared grid-level embeddings may introduce noise rather than useful generalization. In such cases, a single-level, POI-specific embedding that devotes full representational capacity to distinguishing individual locations may be preferable, particularly if there is enough training data to overcome gradient sparsity without sharing. More generally, we believe MiLES is most beneficial when two conditions are met: when spatial proximity is at least partially informative and when the per-location training signal is sparse. These conditions are typical of the online TUL setting but not unique to it.

Furthermore, although MiLES reduces the rate of parameter growth compared to baseline methods (as shown in Table 5), it relies on the dynamic expansion of embedding tables whenever new grid cells or POIs are encountered. In our experiments, pre-allocation sidesteps this issue because the full spatial extent is known, but in a true unbounded stream, embedding tables would grow monotonically with the number of distinct grid cells observed. This means that MiLES may eventually exhaust available memory for a long-running stream over a large or expanding spatial domain. Future work could address this by investigating fixed-budget mechanisms such as embedding pruning or fixed-size hash-based tables.

Our evaluation follows the standard supervised online learning protocol, in which the true user label is assumed to be available immediately following each prediction. In practice, however, feedback may be delayed or incomplete, and the behavior of the model under such conditions remains unexplored. Similarly, although we motivate online learning through non-stationarity, privacy, and resource constraints, we do not empirically evaluate these advantages in isolation. For example, we do not compare online learning against periodic batch retraining with varying retrain intervals. Promising directions for future work include investigating the effectiveness of MiLES under semi-supervised feedback and against incremental batch retraining baselines.

### Broader Impact Statement

By improving the performance of mobility data mining models, the methods presented in this paper have significant broader impacts. While they could enable societal benefits in areas like urban planning and logistics, they also pose privacy risks if misused for surveillance or unwanted tracking. More capable models increase this risk regardless of training paradigm, as higher accuracy enables more reliable re-identification of individuals. Consequently, any real-world deployment should incorporate robust privacy-preserving mechanisms, such as differential privacy (Mir et al., 2013; Yin et al., 2018), or anonymization as a prerequisite for ethical application. At the same time, the online setting central to our work allows trajectory data to be discarded immediately after each model update, eliminating the need to retain sensitive movement records for training purposes.

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

## A Experimental Details and Reproducibility

### A.1 Hyperparameter Values

As described in the main paper, all models and techniques were tuned using the first 5,000 trajectories of the 400-user Foursquare-TKY stream. This protocol was applied identically to all methods to ensure a fair comparison. Table 11 lists the resulting hyperparameter values used in our experiments. For MiLES, $l^{(\max)} = 4$ corresponds to one POI-level embedding and three grid-based sharing levels on Foursquare datasets, and four grid-based levels on GeoLife, which lacks POI information.

### A.2 Computing Infrastructure

- GPU: Nvidia RTX 3090
- CPU: Intel Core i5-9600K
- System Memory: 32 GB DDR4@2133MT/s
- OS: Ubuntu 22.04 LTS
- CUDA: 12.5
- Python: 3.12
- PyTorch: 2.3.1

### A.3 Full Results for All Dataset Variants

The results for all evaluated TUL approaches as well as the dataset-variants with lower user counts are displayed in Table 12. As expected, all models achieved better performance on the lower user-count datasets. In terms of individual models, BiTULER remains the overall most performant model independent of the user count, except for the GeoLife dataset where the GRU-variant of TULER and DeepTUL yielded better results.

The bottom section of Table 12 shows significant performance gains, when substituting the original embedding techniques of the evaluated models with MiLES for both high- and low user-count datasets. The performance benefits at lower user-counts are even higher compared to the larger dataset variants with top-1 accuracy gains of up to 9.79%. The larger benefit for data with fewer users likely stems from the fact that the lower user-count causes the individual users to be more easily identified based on the higher-level embeddings of MiLES.

### A.4 Full Results for Embedding & Replay Methods

The full results comparing MiLES to other embedding techniques as well as general online learning approaches are shown in Table 13. MiLES outperforms all other embedding techniques across all models and datasets. In many cases it even exceeds the performance of replay techniques despite that fact that MiLES adds minimal computational overhead, or reduces overhead, whereas the latter require significantly more computation. It can also be seen that MiLES can be paired with replay techniques to combine their performance benefits. For the more sophisticated TUL models (MainTUL, T3S, TULHOR and TULVAE), a combination of FIFO and MiLES yield the best performance across all metrics and datasets.

Table 11: Tested hyperparameter values for the evaluated models and methods. The selected values are underlined. If different values were selected depending on the used model, multiple are underlined.

| Hyperparameter | Symbol | Tested Values |
|---|---|---|
| Full embedding dim. | $d$ | $\{512, \underline{1024}\}$ |
| Embedding levels | $l^{(\mathrm{max})}$ | $\{3, \underline{4}, 5\}$ |
| Base level grid rows | $h_1^{(\mathrm{max})}$ | $\{100, \underline{200}, \ldots, 500\}$ |
| Level $l$ grid rows | $h_l^{(\mathrm{max})}$ | $\{\underline{2}, 3, 4\}^{-l} \times h_1^{(\mathrm{max})}$ |
| Embedding dim. decay | $\alpha$ | $\{1, \underline{2}\}$ |
| Fourier feature scale | — | $\{1, 2, \underline{4} \ldots, 16\} \times 10^3$ |
| Replay samples/update | — | $\{\underline{1}, 2, 4\}$ |
| Optimizer | — | Adam |
| Learning rate | — | $\{.5, \underline{1}, \underline{2}, \underline{4}, 8\} \times 10^{-4}$ |
| Hidden layers | — | $\{\underline{1}, \underline{2}\}$ |
| Hidden units | — | $\{512, \underline{1024}\}$ |

Table 12: Top-1 accuracy, top-5 accuracy and macro F1 [%], as well as their diffences when using MiLES instead of the default embeddings, for all TUL models on all dataset variants.

| | Dataset | Foursquare-NYC | | | Foursquare-TKY | | | GeoLife | | |
|---|---|---|---|---|---|---|---|---|---|---|
| # Users | Model | Acc@1 | Acc@5 | F1 | Acc@1 | Acc@5 | F1 | Acc@1 | Acc@5 | F1 |
| | | | | | Default | | | | | |
| Low | BiTULER | **69.95** | **75.85** | **68.59** | **73.29** | **84.06** | **72.03** | **39.98** | 74.16 | 32.95 |
| | TULVAE | 69.77 | 75.72 | 68.37 | 66.86 | 77.42 | 64.05 | 39.40 | 73.82 | 31.57 |
| | DeepTUL | 68.93 | 74.77 | 67.62 | 71.65 | 82.47 | 70.42 | 38.44 | 75.15 | **33.42** |
| | MainTUL | 66.28 | 72.41 | 64.86 | 70.69 | 82.30 | 69.28 | 35.89 | 73.02 | 27.87 |
| | T3S | 63.84 | 70.42 | 61.84 | 66.77 | 79.41 | 64.89 | 37.79 | 74.68 | 29.06 |
| | TULHOR | 64.60 | 71.08 | 62.69 | 67.21 | 79.57 | 65.36 | 37.22 | **75.43** | 30.87 |
| High | BiTULER | **60.12** | **67.20** | **57.83** | **63.16** | **74.91** | **61.06** | **37.56** | 70.85 | 26.69 |
| | TULVAE | 59.79 | 66.77 | 57.32 | 55.82 | 66.23 | 51.68 | 37.08 | 70.45 | 25.25 |
| | DeepTUL | 58.72 | 65.48 | 56.60 | 61.17 | 72.49 | 59.10 | 36.32 | **72.64** | **29.82** |
| | MainTUL | 55.67 | 62.61 | 53.01 | 59.53 | 71.89 | 57.09 | 34.00 | 70.26 | 21.76 |
| | T3S | 52.98 | 60.28 | 49.50 | 56.41 | 69.26 | 53.24 | 35.25 | 71.11 | 21.52 |
| | TULHOR | 53.85 | 61.13 | 50.40 | 56.61 | 69.61 | 53.45 | 34.65 | 72.46 | 24.92 |
| | | | | | Δ MiLES | | | | | |
| Low | BiTULER | +2.15 | +3.99 | +2.43 | +1.38 | +2.24 | +1.44 | +8.97 | +6.80 | +8.40 |
| | TULVAE | +2.22 | +4.02 | +2.51 | +2.88 | +3.80 | +3.40 | +8.96 | +6.82 | +8.70 |
| | DeepTUL | +1.67 | +3.54 | +1.98 | +0.80 | +1.95 | +0.76 | +8.93 | +6.15 | +8.55 |
| | MainTUL | +2.17 | +4.67 | +2.29 | +1.25 | +2.54 | +1.08 | +9.11 | +6.65 | +8.68 |
| | T3S | +1.83 | +3.09 | +2.07 | +1.77 | +2.20 | +1.85 | +7.09 | +4.39 | +7.00 |
| | TULHOR | +1.81 | +3.00 | +2.06 | +1.64 | +2.10 | +1.65 | +8.21 | +4.71 | +7.51 |
| High | BiTULER | +1.49 | +3.58 | +1.71 | +1.40 | +2.85 | +1.30 | +8.72 | +6.98 | +6.26 |
| | TULVAE | +1.79 | +3.73 | +2.04 | +3.07 | +4.43 | +3.34 | +8.01 | +6.57 | +4.74 |
| | DeepTUL | +1.06 | +3.04 | +1.26 | +0.84 | +2.37 | +0.77 | +8.59 | +6.31 | +5.97 |
| | MainTUL | +1.44 | +4.33 | +1.52 | +1.62 | +3.45 | +1.45 | +8.19 | +6.18 | +5.50 |
| | T3S | +1.71 | +3.13 | +2.10 | +1.70 | +2.79 | +1.91 | +6.98 | +4.80 | +5.18 |
| | TULHOR | +1.71 | +3.16 | +2.11 | +1.85 | +2.80 | +2.03 | +8.20 | +5.03 | +5.68 |

# B    In-Depth Analysis of MiLES

## B.1    Formal Analysis of the Density-Information Trade-Off

In the main paper, we argue that sharing embeddings creates a trade-off: it increases gradient density and thereby adaptation speed at the cost of informational content. This section provides a formal analysis of this trade-off by quantifying the information loss that occurs when spatial embeddings are shared across multiple locations.

The fundamental principle underlying this analysis is that when distinct locations are mapped to the same embedding, the model loses the ability to distinguish between them. This reduction in discriminability can be quantified as a decrease in Shannon entropy (Shannon, 1948).

Consider a subset $\mathbb{J}$ of location indices that correspond to at least two distinct locations within the same grid cell. We analyze two scenarios: distinct embeddings versus shared embeddings for these locations.

**Distinct Embeddings.**    When each location index $k \in \mathbb{J}$ has its own unique embedding $z(k)$, the contribution of these embeddings to the total information content is measured by their entropy:

$$H_{\text{distinct}} = -\sum_{k \in \mathbb{J}} p(z(k)) \log p(z(k)). \tag{5}$$

Table 13: Top-1 accuracy, top-5 accuracy and macro F1 score for embedding and general online learning techniques for all TUL approaches.

| Dataset | Foursquare-NYC | | | Foursquare-TKY | | | GeoLife | | |
|---|---|---|---|---|---|---|---|---|---|
| Method | Acc@1 | Acc@5 | Macro F1 | Acc@1 | Acc@5 | Macro F1 | Acc@1 | Acc@5 | Macro F1 |
| BiTULER | | | | | | | | | |
| Lookup | 60.12 | 67.20 | 57.83 | 63.16 | 74.91 | 61.06 | 37.56 | 70.85 | 26.69 |
| Linear | 55.75 | 64.45 | 53.08 | 60.63 | 73.45 | 58.20 | 37.28 | 71.13 | 23.33 |
| Fourier | 49.28 | 61.75 | 46.07 | 55.61 | 72.33 | 52.64 | 37.35 | 71.39 | 24.92 |
| MiLES | **61.61** | **70.78** | 59.54 | **64.56** | **77.76** | **62.37** | **46.28** | **77.83** | **32.96** |
| Balanced | 60.04 | 66.95 | 57.95 | 63.08 | 74.56 | 60.89 | 36.91 | 67.83 | 25.28 |
| FIFO | 59.89 | 66.93 | 57.93 | 62.99 | 74.55 | 60.87 | 36.74 | 68.50 | 25.73 |
| FIFO + MiLES | **61.61** | 70.68 | **59.76** | 64.39 | 77.38 | 62.17 | 45.62 | 76.32 | 32.05 |
| DeepTUL | | | | | | | | | |
| Lookup | 58.72 | 65.48 | 56.60 | 61.17 | 72.49 | 59.10 | 36.32 | 72.64 | 29.82 |
| Linear | 55.86 | 64.21 | 53.58 | 59.75 | 72.12 | 57.45 | 35.74 | 72.44 | 26.61 |
| Fourier | 39.51 | 53.45 | 36.29 | 47.04 | 65.62 | 44.07 | 33.79 | 71.23 | 27.38 |
| MiLES | 59.77 | 68.52 | 57.86 | 62.01 | **74.86** | 59.87 | 44.92 | **78.95** | 35.79 |
| Balanced | 59.09 | 65.82 | 57.14 | 61.34 | 72.49 | 59.18 | 34.60 | 65.42 | 26.94 |
| FIFO | 58.86 | 65.65 | 56.91 | 61.20 | 72.47 | 59.09 | 37.89 | 71.62 | 29.75 |
| FIFO + MiLES | **60.00** | **68.70** | **58.16** | **62.21** | 74.77 | **60.02** | **47.01** | 78.53 | **36.09** |
| MainTUL | | | | | | | | | |
| Lookup | 55.67 | 62.61 | 53.01 | 59.53 | 71.89 | 57.09 | 34.00 | 70.26 | 21.76 |
| Linear | 51.48 | 60.34 | 48.19 | 57.59 | 71.04 | 54.67 | 33.48 | 69.98 | 20.25 |
| Fourier | 43.34 | 56.85 | 39.35 | 52.48 | 69.98 | 48.78 | 33.03 | 69.76 | 20.57 |
| MiLES | 57.11 | 66.94 | 54.54 | 61.15 | 75.34 | 58.54 | 42.19 | 76.44 | 27.27 |
| Balanced | 56.34 | 63.55 | 54.26 | 59.61 | 72.02 | 57.06 | 33.01 | 64.27 | 20.32 |
| FIFO | 56.11 | 63.38 | 54.14 | 59.63 | 72.15 | 57.47 | 35.81 | 69.90 | 22.08 |
| FIFO + MiLES | **57.71** | **67.65** | **55.64** | **61.30** | **75.51** | **58.92** | **44.59** | **76.87** | **28.60** |
| T3S | | | | | | | | | |
| Lookup | 52.98 | 60.28 | 49.50 | 56.41 | 69.26 | 53.24 | 35.25 | 71.11 | 21.52 |
| Linear | 39.56 | 47.52 | 35.49 | 38.65 | 52.21 | 34.85 | 33.53 | 69.51 | 19.97 |
| Fourier | 44.68 | 56.18 | 40.87 | 50.81 | 66.98 | 47.31 | 33.79 | 69.82 | 20.59 |
| MiLES | 54.69 | 63.42 | 51.60 | 58.10 | 72.05 | 55.16 | 42.23 | 75.90 | 26.70 |
| Balanced | 54.14 | 61.37 | 51.02 | 56.74 | 69.25 | 53.64 | 39.04 | 72.90 | 25.87 |
| FIFO | 54.71 | 62.06 | 52.26 | 57.43 | 69.85 | 54.94 | 37.48 | 71.68 | 23.38 |
| FIFO + MiLES | **56.10** | **64.67** | **53.77** | **59.05** | **72.30** | **56.54** | **45.12** | **77.23** | **29.01** |
| TULHOR | | | | | | | | | |
| Lookup | 53.85 | 61.13 | 50.40 | 56.61 | 69.61 | 53.45 | 34.65 | 72.46 | 24.92 |
| Linear | 46.90 | 55.26 | 42.85 | 51.77 | 65.17 | 48.29 | 32.53 | 70.22 | 22.22 |
| Fourier | 46.53 | 57.69 | 42.75 | 51.61 | 67.46 | 48.17 | 32.79 | 70.51 | 22.69 |
| MiLES | 55.56 | 64.29 | 52.51 | 58.46 | 72.41 | 55.48 | 42.85 | 77.49 | 30.61 |
| Balanced | 54.41 | 61.71 | 51.15 | 56.98 | 69.58 | 53.77 | 38.36 | 73.92 | 28.41 |
| FIFO | 55.02 | 62.40 | 52.48 | 57.75 | 70.24 | 55.13 | 36.90 | 72.33 | 26.82 |
| FIFO + MiLES | **56.44** | **65.00** | **54.01** | **59.32** | **72.64** | **56.75** | **44.65** | **77.77** | **32.21** |
| TULVAE | | | | | | | | | |
| Lookup | 59.79 | 66.77 | 57.32 | 55.82 | 66.23 | 51.68 | 37.08 | 70.45 | 25.25 |
| Linear | 54.90 | 63.46 | 51.88 | 52.48 | 63.52 | 47.53 | 37.57 | 70.43 | 23.56 |
| Fourier | 48.54 | 60.81 | 44.75 | 49.54 | 63.96 | 43.99 | 36.51 | 70.65 | 23.40 |
| MiLES | 61.57 | 70.50 | 59.36 | 58.89 | 70.66 | 55.01 | 45.10 | 77.02 | 29.99 |
| Balanced | 59.09 | 66.13 | 56.79 | 54.73 | 64.95 | 50.30 | 36.49 | 65.89 | 25.33 |
| FIFO | 59.69 | 66.79 | 57.72 | 57.71 | 68.31 | 54.35 | 39.14 | 71.21 | 27.18 |
| FIFO + MiLES | **61.78** | **70.60** | **59.86** | **60.75** | **72.56** | **57.67** | **48.17** | **78.25** | **33.29** |

**Shared Embeddings.** When all location indices in $\mathbb{J}$ use a single shared embedding $\mathbf{z}^{(\text{share})}$, the probability mass concentrates on this single embedding. The probability of observing $\mathbf{z}^{(\text{share})}$ becomes the sum of the individual probabilities: $p(\mathbf{z}^{(\text{share})}) = \sum_{k \in \mathbb{J}} p(z(k))$. Consequently, the entropy contribution becomes:

$$H_{\text{shared}} = -\sum_{k \in \mathbb{J}} p(z(k)) \log \left( \sum_{k \in \mathbb{J}} p(z(k)) \right). \tag{6}$$

**Information Loss Quantification.** The information loss due to sharing embeddings is given by the entropy difference:

$$H_{\text{shared}} - H_{\text{distinct}} = -\sum_{k \in \mathbb{J}} p(z(k)) \log \left( \frac{\sum_{k \in \mathbb{J}} p(z(k))}{p(z(k))} \right). \tag{7}$$

Since $p(z(k)) > 0$ for all $k \in \mathbb{J}$ and the sum $\sum_{k \in \mathbb{J}} p(z(k)) > p(z(k))$ for any individual term, it follows that

$$\log \left( \frac{\sum_{k \in \mathbb{J}} p(z(k))}{p(z(k))} \right) > 0.$$

As a result $H_{\text{shared}} < H_{\text{distinct}}$.

This entropy difference confirms that sharing embeddings reduces information content. Furthermore, as the number of locations in $\mathbb{J}$ increases, the sum $\sum_{k \in \mathbb{J}} p(z(k))$ grows larger, making the entropy difference more negative and indicating greater information loss when more embeddings are aggregated within shared cells.

## B.2  Ablation of Individual Embedding Levels

To isolate the contribution of different levels of spatial granularity, we evaluated BiTULER models trained with single embedding levels from MiLES. Figure 7 shows the top-5 accuracy over the first 10,000 trajectories.

Initially, the models using coarser, shared embeddings (Levels 2 & 3) adapt more quickly, outperforming the POI-only model (Level 0). After approximately 1,500 trajectories, the higher-specificity Level 0 model achieves better peak accuracy. However, it also exhibits greater sensitivity to distribution shifts (e.g., dips around 4k and 6k samples). The full MiLES model successfully combines the stability of the high-level embeddings with the peak performance of the fine-grained ones.

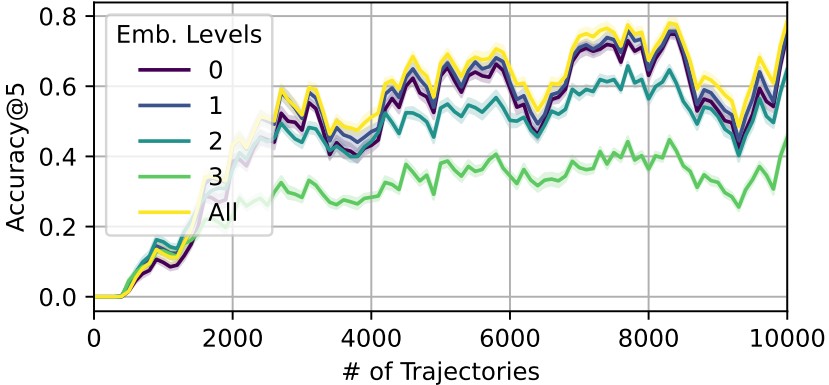

Figure 7: Top-5 accuracy of BiTULER models using individual MiLES embedding levels on Foursquare-NYC.

## B.3  Ablation of Grid Geometry

To validate our choice of a hexagonal grid over a more conventional square grid, we re-ran our evaluations of MiLES using a square grid tiling while keeping all other hyperparameters identical to the main experiments.

As shown in Table 14, the hexagonal grid model consistently outperformed its square-grid counterpart. The performance gap was particularly pronounced on the GeoLife dataset, which does not feature POI information and therefore relies solely on grid-based embeddings. While specifically tuning the hyperparameters of MiLES for a square grid would likely yield improvements, these results empirically support the theoretical advantages of hexagonal grids for better representing spatial proximity in mobility tasks (Ke et al., 2019).

Table 14: Performance comparison of MiLES with hexagonal vs. square grids, using the BiTULER backbone. Results show top-1 accuracy, top-5 accuracy and macro F1. Best results in bold.

| Dataset | Grid Shape | Acc@1 | Acc@5 | F1 |
|---|---|---|---|---|
| Foursquare-NYC | Hexagon | **61.61** | **70.78** | **59.54** |
| | Square | 61.50 | 70.70 | 59.41 |
| Foursquare-TKY | Hexagon | **62.72** | **75.95** | **60.40** |
| | Square | 62.58 | 75.92 | 60.26 |
| GeoLife | Hexagon | **46.28** | **77.83** | **32.96** |
| | Square | 37.01 | 72.82 | 25.39 |

## B.4 Aggregation of Embedding Levels

To validate our choice of concatenation for aggregating MiLES's embedding levels, we compared it against summation. Figure 8 shows the results on the 399-user Foursquare-NYC dataset using BiTULER with fixed level weights.

When summing embeddings, adding more levels degrades top-2 accuracy, suggesting that the less-precise, high-level embeddings interfere with the fine-grained POI embeddings crucial for distinguishing individual users. While summation improves top-5 accuracy in some cases (as coarser features can help identify user groups), the concatenation-based approach consistently achieves the highest performance for both metrics. This empirically supports our design choice for MiLES.

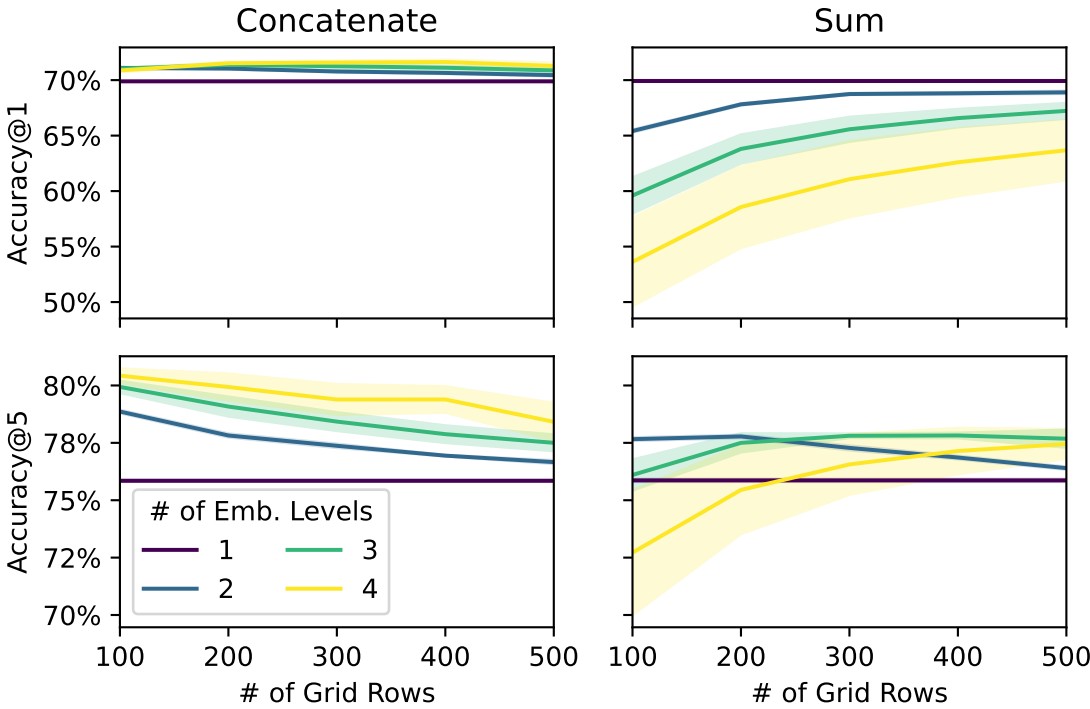

Figure 8: Performance on the 399 user Foursquare-NYC variant, depending on the grid resolution, the number of embedding levels and whether level-specific embeddings were concatenated or summed. BiTULER was used as the classifier and level weights $w_l$ were fixed. Shaded areas represent the $1\sigma$ range.

## C Concept Drift in Trajectory Data

As mentioned above, real-world applications based on mobility data are likely to face changes in the data distribution in the form of concept drift. Such drifts can also be found in the datasets used in this work, and we provide three complementary pieces of descriptive evidence below.

An example for concept drift in trajectory data is the emergence of new POIs. As can be seen in Figure 9, the percentage of visited unique POIs increases quickly at the start of both Foursquare datasets but continues to rise at an almost linear rate throughout both datasets. If a batch learning model were to be trained on the first half of the datasets, it would therefore have encountered only approximately 60% of unique POIs, causing it to struggle interpreting trajectories in the latter half of the datasets, which highlights the need of online learning approaches for mobility mining tasks.

Beyond the emergence of new locations, the distribution over existing POIs also shifts over time. Figure 10 shows the Kullback-Leibler divergence between POI visitation distributions in consecutive trajectory windows on Foursquare-NYC. The divergence increases steadily over the course of the stream, indicating that visitation patterns change substantially even after the initial learning phase. The peak around trajectory 40,000 coincides with a dip in rolling accuracy observed in Figure 2, suggesting that this distributional shift has a tangible impact on model performance.

Figure 11 shows that the composition of active users in GeoLife varies considerably over the data stream. The entropy of user labels within a sliding window fluctuates around a value of 3, until declining sharply after 20000 trajectories have been observed. Such variation changes the effective difficulty of the classification task over time and represents another form of non-stationarity that online models must contend with.

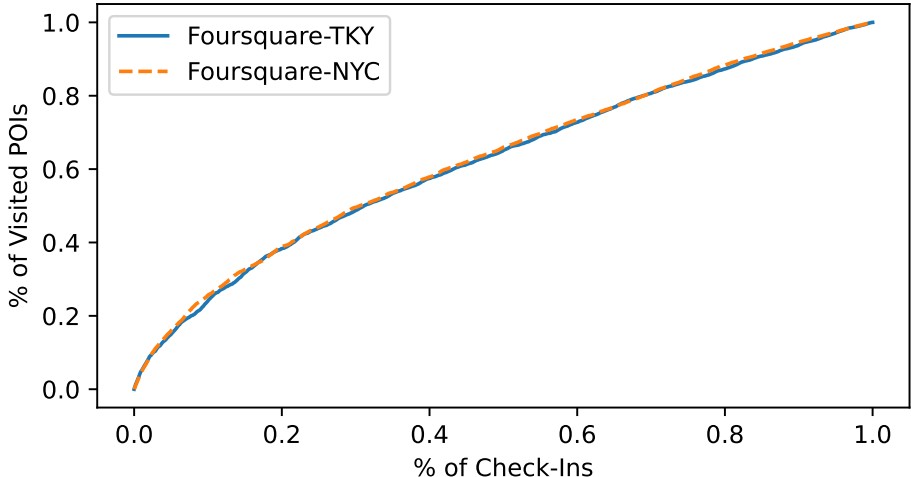

Figure 9: Growth of the observed POI vocabulary relative to the percentage of processed check-ins. The lack of saturation indicates that the model continuously encounters new, unlearned locations throughout the stream.

Figure 12 provides a visual example of concept drift found in the GeoLife dataset. While the trajectories recorded earlier in the collection period and shown in a darker hue are relatively evenly distributed across the area, the later trajectories are much more concentrated around the major transportation axes. This effect could be caused by a change in either user behavior or data collection methodology.

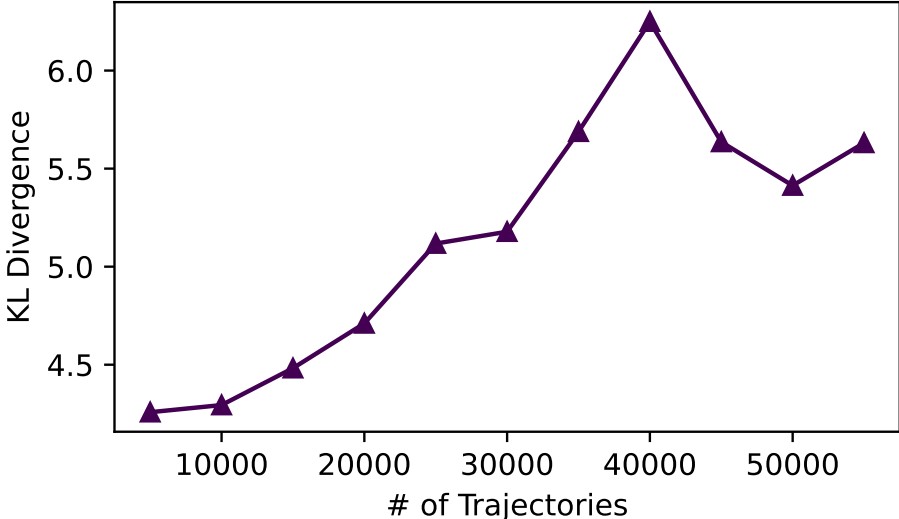

Figure 10: KL divergence between POI visitation distributions in consecutive non-overlapping windows of 5,000 trajectories on Foursquare-NYC. Higher values indicate greater distributional shift between adjacent segments of the stream.

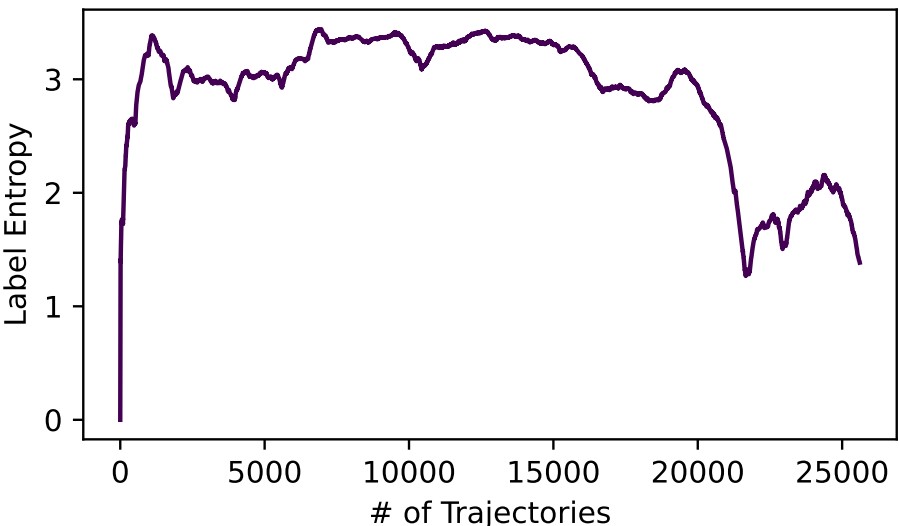

Figure 11: Shannon entropy of user labels within a sliding window of 1,000 trajectories on GeoLife. Fluctuations reflect changes in the number and concentration of active users over the data stream.

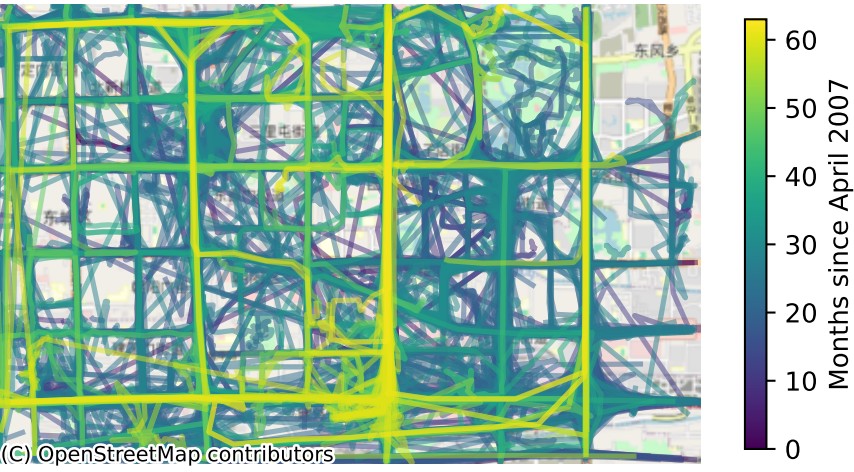

Figure 12: Distribution of trajectories in the GeoLife dataset over time for highly frequented area. Trajectories are color-coded according based on their time of occurrence.

