# OpenReview forum: "Multi-Level Spatial Embedding Sharing for Enhanced Online Trajectory-User Linking"
_TMLR — Rejected by TMLR_

### Review · Reviewer_MBJQ · 2026-03-07

**Summary Of Contributions:**

This paper explores the Trajectory-User Linking (TUL) task within an online learning framework, which the authors claim is a neglected but critical setting. To address the challenges of streaming data and distributional shifts (concept drift), they propose MiLES (Multi-Level Spatial Embedding Sharing). MiLES employs a hierarchical grid-based embedding approach that shares representations across multiple spatial scales and introduces learnable weights to balance their influence. The authors evaluate MiLES on several TUL models and datasets, reporting improvements in Acc@1, Acc@5, and F1-score. They also discuss memory efficiency and robustness to concept drift.

**Audience:**

Yes

**Audience Explanation:**

As currently framed, the paper is heavily tilted toward the Data Mining (GIS/Mobility) community. For TMLR's broader machine learning audience, the interest would lie in how spatial constraints affect representation learning in non-stationary environments. However, because the methodological novelty is thin and the online learning setting is not rigorously justified, its appeal to ML researchers outside of the TUL niche is likely limited.

**Broader Impact Concerns:**

The paper discusses privacy at the end.

**Claims And Evidence:**

No

**Claims Explanation:**

The primary concern is the weak coupling between the proposed MiLES method and the online learning setting. While the authors claim MiLES is specifically designed for online TUL, the evidence does not convincingly show a "special gain" that is unique to the online environment.

- **Lack of Synergy between MiLES and Online Learning**:  Tt is not necessary to combine MiLES and online learning. The authors claim they propose MiLES for online learning. However, the performance improvements reported (e.g., Table 4 ) appear to stem from the inherently superior representational power of multi-level spatial embeddings rather than a specific enhancement of the online adaptation process.

- **Insufficient Evidence for "Faster Adaptation"**: The authors argue that MiLES enables "faster adaptation" and show a "steeper slope" during recovery in Figure 2. However, a closer look at Figure 2  reveals that MiLES maintains a consistent performance offset over the default model both before and after the concept drift. This suggests that the gain is a static improvement in base accuracy rather than a dynamic improvement in the rate of adaptation. A true online-specific gain should demonstrate that the model converges to its peak accuracy significantly faster than the baseline when starting from scratch or after a distribution shift, which is not clearly evident here.

- **Weak Motivation for the Online Setting**: The justification for the online learning requirement is insufficient.
  - **The Labeling Paradox**: In a real-world TUL stream, who provides the ground-truth user labels for online updates? If the user identity is already known for the update, the primary goal of TUL is defeated. The authors should clarify the practical scenario: is this a "test-time adaptation" or "continual learning with a teacher"?
  - **Scalability vs. Necessity**: TUL datasets are typically not so large that they cannot be stored or re-trained periodically offline. The authors fail to demonstrate a scenario where "full training" is computationally impossible but "online update" is mandatory.

**Requested Changes:**

- **Structure and Scope**: The authors mention that "destination prediction" is a main contribution, yet it is relegated to the Appendix to meet a self-imposed (and non-existent for TMLR) page limit. If it is a core claim, it must be in the main text. Note that TMLR has no strict page limit; authors are encouraged to be concise but complete.

- **Clarification of Online Setting**: Provide a rigorous definition of the online learning protocol. Specifically, address the "labeling" issue: how does the model receive feedback in a streaming TUL application?

---

> ### Author Response · Authors · 2026-03-20
>
> We would like to thank the reviewer for their detailed comments. Their feedback prompted us to conduct new experiments, which we believe have strengthened the paper.
>
> **Lack of Synergy between MiLES and Online Learning**
>
> We thank the reviewer for this important observation. We agree that MiLES provides a general representational benefit that is not exclusive to online learning, and we have revised our claims accordingly. To quantify the degree to which MiLES's benefits are setting-specific, we conducted new experiments comparing the performance gains of MiLES over the baseline in both batch and online evaluation modes (see Subsection 6.1). The batch setting uses 20 training epochs with hyperparameters tuned for batch learning to ensure a fair comparison.
>
> The results confirm that MiLES improves performance in both settings, but the gains in top-5 accuracy and macro F1 score are consistently larger in the online regime. The difference is the largest on GeoLife, where MiLES yields a top-1 accuracy improvement of +22.10% in the online setting compared to +4.81% in batch learning.
> We attribute this to the greater severity of gradient sparsity in the online regime.
> Since each trajectory is observed only once, POI embeddings receive far fewer gradient updates than in batch training, where multiple epochs can improve representations for rarely visited locations.
> This effect is amplified on GeoLife, where trajectories are less spatially diverse and a larger fraction of locations remains underrepresented at any point in the stream.
> Additionally, MiLES's learnable level weights (Figure 5) enable the model to adjust the relative contribution of each embedding level dynamically throughout the stream, which we believe is more relevant in a non-stationary online setting.
>
> We have revised the introduction to clearly state that MiLES is not exclusively beneficial in online learning scenarios but that its advantages are more pronounced in such settings (Section 1, paragraph 5).
>
> **Insufficient Evidence for "Faster Adaptation"**
>
> We agree that our original framing overstated the adaptation speed claim. We have revised the paper in several ways:
>
> First, we replaced references to "adaptation speed" with the term "gradient density", which directly describes the property that MiLES's shared embeddings provide (Section 3.1).
>
> Second, we revised the interpretation of the concept drift experiment (Figure 2).
>
> Third, we now point to Figure 6 as supporting evidence that coarser grid-based embedding levels gain accuracy more quickly at the start of the stream, consistent with their reduced sparsity.
>
> **Clarification of Online Setting**
>
> We have added an explicit statement of the feedback assumption to the preliminaries (Section 2), clarifying that we follow the standard supervised online learning protocol where the true user label becomes available after each prediction. We provide a concrete motivating example and acknowledge that settings with delayed or incomplete feedback are a promising direction for future work. The limitations section now also explicitly discusses this assumption and its implications.
>
> **Scalability vs. Necessity**
>
> We agree that most publicly available trajectory datasets are small enough for periodic batch retraining. We have revised the introduction to clarify that our motivation for studying online TUL is not primarily computational necessity but rather the combination of (i) a principled framework for non-stationary environments, (ii) privacy advantages from immediate data disposal, and (iii) relevance to resource-constrained on-device deployment.
>
> **Structure and Scope**
>
> We thank the reviewer for pointing out this oversight. We were mistaken about TMLR's policy regarding the length of submissions and have moved the destination prediction evaluation from the appendix into the main text (Section 7).

---

> > ### Comment · Reviewer_MBJQ · 2026-03-25
> >
> > We thank the authors for their thorough revision and the new experiments in Section 6.1. The comparison between batch and online gains is informative, and the revised claims are more careful than the original submission. However, several concerns remain.
> >
> > **The online setting still feels unnatural for TUL.**
> >
> > While we appreciate the added clarification in Section 2 regarding the supervised online learning protocol, the labeling paradox has not been adequately resolved. The authors now explicitly assume that ground-truth user labels are available immediately after each prediction and offer ride-hailing as a motivating example. However, this example is quite narrow, and in the broader set of TUL applications the authors themselves cite — disease control, law enforcement, location-based recommendations — it is unclear when or why a true label would be available in a streaming fashion. If the deployment scenario is restricted to cases where a "teacher" provides labels after each prediction, the authors should be upfront about how limited this class of applications is, rather than motivating the paper with the full breadth of TUL use cases.
> >
> > **The necessity of online learning remains insufficiently established.**
> >
> > The revised motivation now rests on three pillars: (i) a principled framework for non-stationarity, (ii) privacy advantages, and (iii) on-device deployment. We find each of these somewhat thin on its own. Regarding (i), calling online learning "principled" is not a substitute for demonstrating that it is needed. The natural competing baseline is periodic batch retraining at varying intervals, which is what most deployed systems actually use. Section 6.1 compares full batch training against online learning but does not test this middle ground. If retraining every few thousand trajectories achieves comparable performance to continuous online updates while being simpler to implement, the case for the online setting weakens considerably. Regarding (ii), the privacy argument is not only unsupported by any experiment or formal analysis, but also sits in tension with the task itself. TUL aims to de-anonymize trajectories, and an online model that continuously adapts to evolving mobility patterns could arguably pose greater privacy risks than a static batch model. Regarding (iii), on-device deployment is plausible but also unsubstantiated — no resource constraints are measured or simulated.
> >
> > **The contribution has shifted substantially.**
> >
> >  We acknowledge the authors' intellectual honesty in revising their claims, but this revision has consequences for how the paper should be evaluated. The original submission positioned MiLES as a method designed for online TUL, with the online setting as a core and novel contribution. The revision reframes MiLES as a general-purpose multi-scale embedding technique that happens to be more beneficial in online settings. This is a weaker and fundamentally different claim. What remains is a competent embedding technique — multi-level grid sharing with learnable weights and variable dimensionality — evaluated in a setting whose practical relevance has not been convincingly argued. The Section 6.1 results, while useful, actually reinforce this reading: MiLES helps in batch learning too, meaning the online framing is not essential to the contribution.
> >
> > We suggest the authors consider one of two paths: either (a) provide substantially stronger justification for the online setting, including a comparison against periodic retraining baselines and a concrete, well-developed application scenario where streaming labels are realistic, or (b) reframe the paper around MiLES as a general spatial embedding technique and evaluate it more thoroughly in both batch and online settings, treating the online regime as one of several evaluation conditions rather than the central motivation.

---

> > > ### Author Response · Authors · 2026-03-26
> > >
> > > We thank the reviewer for their continued constructive engagement and for the two concrete paths suggested. We address each concern below and hope to demonstrate that the revised paper already addresses the core of these suggestions.
> > >
> > > **Supervision Assumption**
> > >
> > > The fully-labeled stream protocol we adopt is the most common evaluation paradigm for stream learning algorithms (Aguiar et al., 2024). As Bifet et al. (2023) note, this model "is rightly criticized by practitioners as too simple, because it ignores the very real problem of delayed and missing label feedback," but "is however quite useful for comparing learning algorithms in a clean way." We adopt it for precisely this reason: supervised evaluation is arguably the cleanest protocol for isolating the effect of an embedding technique, which is the focus of our contribution. We acknowledge the limitation explicitly in the revised limitations section.
> > >
> > > That said, several realistic scenarios do provide streaming labels. In fleet management, a driver's identity is confirmed when they scan a badge at a delivery point. In continuous authentication for ride-hailing, verification challenges triggered by mismatching predictions provide immediate labels. Our destination prediction experiments (Section 7) further demonstrate applicability to a task where labels arise naturally as the final location of each trajectory.
> > >
> > > More broadly, MiLES operates at the embedding level and is agnostic to the supervision paradigm. We chose supervised evaluation because it is the most well-established protocol, not because MiLES requires it. Semi-supervised or self-supervised online evaluation is a promising direction for future work.
> > >
> > > **Online Learning vs. Periodic Retraining**
> > >
> > > We agree that periodic retraining can be a practical alternative and have acknowledged this in the revised paper. We note that online learning and batch learning represent the two extremes of a retraining-interval spectrum. Batch learning trains over all data at once, while online learning updates after every observation. The new experiment added to Section 6 evaluates both extremes and shows that MiLES provides gains at both ends of the spectrum, with consistently larger gains at the online end.
> > >
> > > We agree that evaluating intermediate retraining intervals would be informative. However, this would shift the paper's focus from the embedding contribution to a study of retraining schedules, introducing deployment-specific hyperparameters (interval length, fine-tuning vs. retraining from scratch, data retention policy) that are orthogonal to MiLES itself. We consider this complementary and valuable future work, but believe it is beyond the scope of the current paper's contribution.
> > >
> > > **Nature of the Contribution**
> > >
> > > We understand the reviewer's concern. However, we believe the revised framing of MiLES as a general multi-scale embedding technique with a disproportionate benefit in online settings is a more precise claim rather than a weaker one. We would like to offer two observations:
> > >
> > > First, we view MiLES's effectiveness in batch learning as complementary to its benefits in online learning, rather than as conflicting with them. In our view, generalization across settings is a desirable property. The relevant question is whether there is setting-specific synergy, which our new results confirm. MiLES improves batch performance by +4–8% but online performance by up to +22% on GeoLife. This asymmetry is supported by the explanation that gradient sparsity is more severe under single-pass learning and cannot be compensated through multiple epochs.
> > >
> > > Second, we note that the reviewer's suggested path (b) of framing MiLES as a general technique evaluated across settings is close to what the revised paper already does. We evaluate MiLES in both batch and online regimes, demonstrate gains in both, provide an explanation for the asymmetry, and include destination prediction as a second task to the main paper, demonstrating generality. The online setting remains the primary motivation because it is both the more challenging regime and the one where MiLES's inductive bias provides the greatest empirical benefit.
> > >
> > > **Privacy**
> > >
> > > Regarding privacy, we note that online learning enables immediate discarding of trajectory data after each update, reducing the attack surface compared to retaining historical data for periodic retraining.
> > >
> > > ---
> > >
> > > Aguiar, Gabriel, et al. "A survey on learning from imbalanced data streams: taxonomy, challenges, empirical study, and reproducible experimental framework." Machine learning 113.7 (2024).
> > >
> > > Bifet, Albert, et al. "Machine learning for data streams: with practical examples in MOA." MIT press, (2023).

---

> > > > ### Comment · Reviewer_MBJQ · 2026-04-01
> > > >
> > > > We thank the authors for their continued engagement.
> > > >
> > > > We find the supervision assumption defense reasonable — the Bifet et al. (2023) citation and the additional application examples help clarify the evaluation protocol's standing. We accept this point.
> > > >
> > > > However, the response regarding periodic retraining is not convincing. The authors characterize this comparison as "beyond scope," but the online setting is a central part of the paper's motivation and framing. If the authors wish to claim that online learning is the right framework for TUL, demonstrating its advantage over the practical middle ground of periodic retraining is not orthogonal — it is directly relevant to the paper's central argument.
> > > >
> > > > Regarding privacy, the authors did not address the tension we raised. Our concern was not about data retention but about the fact that TUL is itself a de-anonymization task. To give a concrete example: consider a person who has relocated for safety reasons. A stale batch model might lose the ability to identify them as their routines change, but an online TUL model that continuously adapts would learn their new patterns and maintain its ability to link their trajectories to their identity — effectively enabling persistent tracking. The training data has been discarded, but the model can still answer "where is this person right now?" The privacy harm lies in the model's capability, not in the stored data. Framing online learning as privacy-preserving in this context is difficult to justify.
> > > >
> > > > We maintain that the paper would benefit from either strengthening the online motivation with the suggested experiments or repositioning MiLES as a general embedding contribution with the online setting as one evaluation condition among several.

---

> ### Author Response · Authors · 2026-04-02
>
> We thank the reviewer for accepting our defense of the supervision protocol and for continuing the discussion on the remaining points.
>
> **Privacy**
>
> The reviewer raises a valid concern about model capability enabling persistent re-identification, which we acknowledge in our broader impact statement.
> However, this risk is a function of model accuracy, not the training paradigm.
> A batch- or periodically retrained model with equivalent performance poses the same risk.
> Our privacy motivation refers to a different property: online learning allows training data to be discarded immediately after each update, eliminating the risk of stored trajectory data being breached or misused.
> We have clarified this distinction in the revised text to avoid conflating the two concerns.
>
> **Periodic Retraining**
>
> We want to clarify an important point: the paper does not claim that online learning is superior to periodic retraining or that it is *the* right framework for TUL.
> It claims that online learning is a viable and previously unstudied framework, and that MiLES is particularly beneficial in this setting.
> We explicitly acknowledge periodic retraining as a practical alternative and list its investigation as future work.
>
> Among the reasons why we consider online learning worth studying in its own right are lower memory requirements, no dependence on buffer sizes or update schedules and immediate adaptation to new data.
> We note that the broader stream learning literature routinely evaluates algorithms under the online protocol without requiring a comparison against periodic retraining as a prerequisite (e.g., Gama et al., 2014; Lu et al., 2018; Bifet et al., 2023), treating them as distinct paradigms whose relative merits depend on the deployment context.
>
> Given this framing, we respectfully maintain that a comparison against periodic retraining, while informative, is not required to support the paper's claims.
>
> ---
>
> Bifet, Albert, et al. "Machine learning for data streams: with practical examples in MOA." MIT press, 2023.
>
> Gama, João, et al. "A survey on concept drift adaptation." ACM computing surveys (CSUR) 46.4 (2014): 1-37.
>
> Lu, Jie, et al. "Learning under concept drift: A review." IEEE transactions on knowledge and data engineering 31.12 (2018): 2346-2363.

---

### Review · Reviewer_uB57 · 2026-03-07

**Summary Of Contributions:**

This paper studies trajectory-user linking in an online learning setting, where data arrives sequentially and the model must predict first and then update. The paper argues that this setting is more realistic for many mobility applications than the standard batch setting, especially under distribution shift and the continuous arrival of new POIs. On top of this evaluation setting, the paper proposes MiLES, a multi-level spatial embedding sharing method that combines POI-level and multi-scale grid-level spatial embeddings through concatenation and learnable per-level weights. The main idea is to balance fast adaptation from shared coarse spatial embeddings with fine-grained discrimination from specific embeddings.

The empirical study is broad. The authors first adapt several established TUL models to online evaluation and compare their performance. They then replace the original embedding modules with MiLES and report consistent gains across Foursquare-NYC, Foursquare-TKY, and GeoLife, with especially strong gains on GeoLife. The paper also includes comparisons against alternative embedding choices, replay methods, adaptive optimizers, component ablations, a simulated concept-drift experiment, and an additional destination-prediction study to support the claim that MiLES is not specific to TUL alone.

Strengths:
1. The paper addresses a relevant and underexplored online setting. The proposed method is simple and modular.
2. The parameter reduction and runtime discussion are also useful, since MiLES is presented not only as an accuracy improvement but also as a more efficient embedding design.

Weaknesses:
1. Technical novelty is moderate.
2. The evidence is almost entirely empirical, with limited theoretical depth beyond the entropy-style discussion in the appendix.
3. The tuning protocol may raise some concerns because hyperparameters are selected on one specific stream segment and then transferred to all datasets and models.

**Additional Comments:**

NAN

**Audience:**

Yes

**Audience Explanation:**

I expect there would be clear interest from readers working on mobility modeling, online learning, spatiotemporal representation learning, and streaming ML systems.

**Broader Impact Concerns:**

The paper already includes a broader impact statement and correctly notes privacy and surveillance risks in mobility modeling, especially in an online setting where models can continuously adapt to individuals’ movement patterns. I think this is the main ethical issue and it is relevant.

**Claims And Evidence:**

Yes

**Claims Explanation:**

Overall, the empirical evidence is convincing and generally supports the paper’s main claims. The authors clearly define the online evaluation protocol using a prequential test-then-train setup, evaluate multiple TUL backbones, and show consistent improvements after replacing the default embedding modules with MiLES. The gains are reported on several datasets and metrics, and the paper also includes efficiency measurements, ablations, and significance tests. This gives the paper a solid empirical basis.

**Requested Changes:**

1. The paper should sharpen its positioning on novelty. The method is effective and well motivated, but the main novelty seems to lie in adapting multi-level spatial sharing to the online TUL setting, rather than in introducing a fully new modeling principle. I encourage the authors to state this more precisely and to make the distinction from prior grid-based sharing and multi-scale spatial encoding methods clearer in the introduction and related work. (critical)

2. The claims about concept drift should be stated more carefully. The paper gives a reasonable motivation for drift in mobility streams and includes both descriptive evidence and a controlled experiment, but some wording in the paper reads broader than what is directly established by the experiments. I suggest narrowing the claim to what is actually shown, or adding a clearer discussion of the limits of the current drift evidence. (critical)

3. The hyperparameter tuning protocol should be discussed in more depth. The appendix states that all models and techniques were tuned on the first 5,000 trajectories of the 400-user Foursquare-TKY stream and then reused across experiments. This is a reasonable choice for consistency, but it may also bias conclusions toward one stream segment or one dataset. The paper would be stronger if the authors justify this design more explicitly and comment on how sensitive the findings are to this protocol. (critical)

4. The paper should make the implementation setting clearer regarding pre-allocation versus true dynamic expansion. The method is presented as supporting dynamically growing embedding tables for unseen locations, but the experiments use pre-allocated tables when the full index range is known in advance. This does not invalidate the results, but the practical implications for a true unbounded stream, memory growth, and deployment should be discussed more directly in the main text. (critical)

5. The limitations section could be expanded a bit. In particular, the current text already notes that MiLES may be weaker for tasks needing very fine distinctions between nearby locations and that memory may still grow over time. It would be useful to connect these limitations more explicitly to the main claims and to discuss when a simpler single-level embedding might still be preferable. (non-critical)

6. The paper would benefit from a short discussion of statistical stability over time in the online stream, beyond final averaged metrics. Since the main motivation is online adaptation, it would be useful to emphasize stream-wise behavior and not only aggregate end results, especially in settings where user composition or POI coverage changes over time. (non-critical)

---

> ### Author Response · Authors · 2026-03-20
>
> We thank the reviewer for their thorough and constructive feedback, which has helped us improve the clarity and rigor of the paper.
>
> **Novelty Positioning**
>
> We agree and have revised the paper accordingly. The introduction now explicitly acknowledges grid-based and multi-scale spatial encoding methods and positions MiLES as adapting this principle to online learning constraints.
> The related work now more clearly identifies the prior methods, including T3S, TULHOR, the newly added Space2Vec (Mai et al., 2019) and Fourier features.
> The contributions list has been updated to reflect this framing.
>
> **Concept Drift Claims**
>
> We appreciate this observation and have clarified our claims throughout the paper. We also expanded the appendix with two new analyses (KL divergence of POI distributions on Foursquare-NYC and label entropy on GeoLife), which show that distributional shifts are present in the unmodified datasets as well, not only in our controlled experiment.
> Since MiLES consistently improves performance across these datasets, its benefits are at least maintained under the types of drift naturally present in them.
>
> **Hyperparameter Tuning Protocol**
>
> This is a valid concern. We have added an explicit justification to the experimental setup: the protocol reflects realistic online deployment constraints.
> We note that (1) hyperparameters were tuned on Foursquare-TKY but improvements hold across all three datasets, (2) the sensitivity analysis on Foursquare-NYC shows robustness across grid resolutions and embedding levels, and (3) dataset-specific tuning would be expected to only strengthen MiLES's results, making the current protocol conservative.
>
> **Pre-allocation vs. Dynamic Expansion**
>
> We added clarification in the approach section discussing the practical implications of dynamic expansion in deployment, and in the limitations section connecting our pre-allocated experimental setup to the unbounded-stream case.
>
> **Limitations Expansion**
>
> We added hypotheses regarding when MiLES is most and least beneficial: it helps most when spatial proximity is informative and per-location training signal is scarce, and may introduce noise when neighboring locations serve fundamentally different roles. In such cases, a single-level POI-specific embedding may be preferable.
>
> **Statistical Stability over Time**
>
> We appreciate this suggestion and have added a new figure showing rolling top-1 accuracy over the stream for Foursquare-NYC and GeoLife with confidence bands.
> On Foursquare-NYC, MiLES maintains a consistent advantage throughout, with the gap widening slightly during a period of elevated distributional shift. On GeoLife, the advantage grows over the course of the stream.

---

### Review · Reviewer_2R5p · 2026-03-16

**Summary Of Contributions:**

This paper studies trajectory-user linking (TUL) in an online / prequential setting and argues that standard POI lookup embeddings are too sparse for fast adaptation under streaming data and concept drift. To address this, it proposes MiLES, a drop-in multi-level spatial embedding module that combines POI/grid embeddings at multiple spatial resolutions using per-level learnable weights, while also allowing dynamic expansion to unseen locations. Empirically, the paper evaluates MiLES across several TUL backbones on Foursquare-NYC, Foursquare-TKY, and GeoLife, reports consistent gains over default embeddings, analyzes parameter count and runtime, includes ablations and a simulated drift experiment, and adds an appendix showing transfer to destination prediction.

**Audience:**

Yes

**Audience Explanation:**

The paper does provide meaningful supporting evidence for the main empirical claim that MiLES often improves online TUL performance.

The reported improvements are consistent across multiple backbones, with especially large gains on GeoLife, and the paper also shows reduced parameter counts and competitive runtime, plus complementary gains when MiLES is combined with FIFO replay.

The appendix further suggests that the idea is not limited to TUL, since destination prediction also improves in many settings.

**Broader Impact Concerns:**

N/A in my opinion.

**Claims And Evidence:**

Yes

**Claims Explanation:**

Overall, I found the direction promising and the empirical story broadly positive.

The paper addresses a relevant setting that appears less explored than standard batch TUL, and the core idea is simple, modular, and potentially useful beyond the specific task studied here.

At the same time, I do not think the current version is yet fully convincing as written, mainly because some core methodological details remain under-specified and several of the stronger experimental claims need clearer justification.

**Requested Changes:**

Although the paper has several positive aspects on TUL, I do not think the evidence is fully clear or fully convincing yet.

First, the method section is not sufficiently self-contained. The paper defines online TUL with a generic loss in Eq. (2), but MiLES itself is only specified as an embedding construction for a single check-in. In particular, the check-in is defined as $c=(t,l)$, yet the role of timestamp $t$ in the end-to-end training pipeline remains unclear from the MiLES description itself. The paper later mentions that timestamps are included and that hour-specific lookup embeddings are used, but the final interaction between MiLES, temporal encoding, and each downstream backbone is still not described concretely enough for the reader to fully understand or reproduce the full training pipeline from the main method section alone.

Second, the description of the learnable weights is somewhat overstated. In the paper, these weights are described as an “attention mechanism” that dynamically adjusts the influence of each level based on the data stream. But from Eq. (4) and Algorithm 1, they appear to be global learnable scalar multipliers for each level, rather than input-dependent attention weights. That does not make the method invalid, but I think the current presentation is somewhat stronger than what is actually implemented.


Third, the experimental protocol needs stronger justification across datasets. Hyperparameters are tuned on the first 5,000 trajectories of the 400-user Foursquare-TKY stream and then reused across all datasets and methods. This is a fair uniform protocol, but it is not obvious that it is sufficiently representative given the large differences between the Foursquare datasets and GeoLife, especially since GeoLife lacks POI identifiers entirely. The paper itself later relies on dataset-specific explanations for why some components matter more on some datasets than others, but those explanations would be more convincing if the paper included stronger dataset characterization and/or sensitivity analysis.

Fourth, the concept-drift discussion is suggestive but under-quantified in the main paper. The simulated drift experiment is interesting: the eastern half of Foursquare-NYC is withheld from the first 30,000 trajectories, and MiLES appears to recover faster afterward. But the paper mainly presents this visually, without clearly quantifying the drift magnitude, the proportion of unseen POIs introduced by the intervention, minimum post-drift performance, or recovery speed. Appendix D helps motivate why drift matters by showing continuing POI vocabulary growth and a GeoLife visualization, but even there the discussion remains more qualitative than quantitative.

Finally, parts of the statistical analysis are hard to interpret. Table 5 reports one-sided Wilcoxon signed-rank tests and says that “both variants of each dataset” were used to compute the statistics, but the effective sample unit is not fully clear from the write-up. I would like the paper to state more explicitly what the paired observations are and how the test was constructed.

---

> ### Author Response · Authors · 2026-03-20
>
> We thank the reviewer for their careful reading and specific suggestions, which helped us refine our claims and improve the presentation of our embedding approach.
>
> **Method Section Self-Containedness**
>
> We agree that the method section should describe the full pipeline, not only the per-check-in embedding construction. We have added a paragraph at the end of Section 3 that describes how MiLES embeds a full trajectory. Each check-in's location is passed through MiLES, concatenated with an hour-of-day lookup embedding, and fed to the backbone.
>
> **Learnable Weights Description**
>
> The reviewer is correct that the level weights are global learned scalars, not input-dependent attention weights. We have replaced all references to "attention mechanism" and "level-attention weights" with "learnable level weighting" / "weighting parameters" throughout the paper.
>
> **Hyperparameter Tuning Protocol**
>
> We acknowledge that transferring hyperparameters from Foursquare-TKY to the other datasets, particularly GeoLife, may lead to suboptimal configurations.
> However, we note that this protocol is applied uniformly to all models and methods, including the baselines.
> Therefore, any disadvantage from cross-dataset transfer likely affects all configurations and does not explicitly favor MiLES.
> The protocol also reflects realistic online deployment constraints, where labeled data from the exact target stream may not be available for tuning prior to deployment.
> Additionally, the newly added experiment comparing the impact on MiLES in online learning vs. in batch learning (see Section 6.1) uses a separately tuned hyperparameter configuration, under which MiLES's gains remain comparable in magnitude, suggesting the results are not sensitive to the specific hyperparameter setting. We have revised the experimental setup to make this reasoning explicit.
>
> **Concept Drift Quantification**
>
> We agree that the drift experiment required more rigorous quantification. We have added two new tables. Table 3 quantifies the novelty introduced by the simulated drift at each embedding level: 36.8% of check-ins in the first 1,000 post-drift trajectories reference POIs entirely absent from the pre-drift stream, while this fraction drops below 25% at coarser grid levels. This directly illustrates the mechanism behind MiLES's advantage. Table 4 reports post-drift performance, whith the main finding being that MiLES maintains a higher performance floor (39.4% vs. 36.7% top-1; 46.9% vs. 42.7% top-5). The primary benefit of MiLES under drift therefore appears to be a higher baseline performance. We have modified our analysis of the results to reflect this.
>
> **Statistical Test Clarity**
>
> We have clarified the test construction in both the body text and the table caption. Each paired observation is the metric difference (full MiLES minus ablated) from a single prequential evaluation run, identified by a unique (dataset variant, random seed) combination. With 2 variants and 10 seeds per dataset, each test is based on N = 20 paired observations. The one-sided Wilcoxon signed-rank test evaluates whether these differences are systematically positive against a null hypothesis that their distribution is symmetric about zero.

---

### Decision · Action_Editor_WSby · 2026-04-21

**Recommendation:** Reject

**Audience:**

Yes

**Audience Explanation:**

Trajectory-User Linking is an important task. The paper would attract readers working on this topic.

**Claims And Evidence:**

No

**Claims Explanation:**

Reviewers generally agree that the claims are accurate and convincing. However, all agree that the novelty claims can be further improved, especially justifying why the online setting is the appropriate formulation of the problem. This requires significant rewriting of the paper instead of minor revision. Therefore we would have to reject the paper in the current revision and encourage the authors to revise the paper based on the detailed suggestions from the reviewers.

**Resubmission Of Major Revision:**

The authors may consider submitting a major revision at a later time.